# STARK: Benchmarking LLM Retrieval on Textual and Relational Knowledge Bases

**Shirley Wu**[*§], **Shiyu Zhao**[*§], **Michihiro Yasunaga**[§], **Kexin Huang**[§], **Kaidi Cao**[§], **Qian Huang**[§],
**Vassilis N. Ioannidis**[†], **Karthik Subbian**[†], **James Zou**[‡§], **Jure Leskovec**[‡§]

[§]Department of Computer Science, Stanford University   [†]Amazon

https://stark.stanford.edu/

## Abstract

Answering real-world complex queries, such as complex product search, often requires accurate retrieval from semi-structured knowledge bases that involve blend of unstructured (*e.g.,* textual descriptions of products) and structured (*e.g.,* entity relations of products) information. However, many previous works studied textual and relational retrieval tasks as separate topics. To address the gap, we develop STARK, a large-scale Semi-structure retrieval benchmark on Textual and Relational Knowledge Bases. Our benchmark covers three domains: product search, academic paper search, and queries in precision medicine. We design a novel pipeline to synthesize realistic user queries that integrate diverse relational information and complex textual properties, together with their ground-truth answers (items). We conduct rigorous human evaluation to validate the quality of our synthesized queries. We further enhance the benchmark with high-quality human-generated queries to provide an authentic reference. STARK serves as a comprehensive testbed for evaluating the performance of retrieval systems driven by large language models (LLMs). Our experiments suggest that STARK presents significant challenges to the current retrieval and LLM systems, highlighting the need for more capable semi-structured retrieval systems.

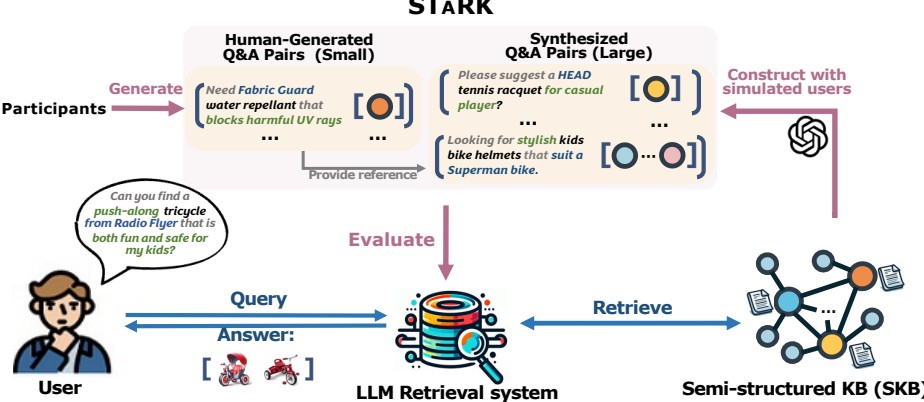

Figure 1: STARK features queries on Semi-structured Knowledge Base (SKB) with textual and relational knowledge, with node entities as ground-truth answers. STARK consists of synthesized queries simulating user interactions with a SKB and human-generated queries which provide an authentic reference. It evaluates LLM retrieval systems' performance in providing accurate responses.

---

[*‡]Equal first-author / senior contribution.

Correspondence: {shirwu, jamesz, jure}@cs.stanford.edu.

38th Conference on Neural Information Processing Systems (NeurIPS 2024) Track on Datasets and Benchmarks.

| | Example query | Title of ground-truth items(s) |
|---|---|---|
| STARK-AMAZON | *Looking for durable Dart World brand dart flights that resist easy tearing. Any recommendations?* | \<Amazon Standard Flights\> \<Dart World Broken Glass Flight\> (+12 more) |
| | *What are recommended scuba diving weights for experienced divers that would fit well with my Gorilla PRO XL waterproof bag?* | \<Sea Pearls Vinyl Coated Lace Thru Weight\> |
| STARK-MAG | *Search publications by Hao-Sheng Zeng on non-Markovian dynamics.* | \<Distribution of non-Markovian intervals...\> \<Comparison between non-Markovian...\> |
| | *What are some nanofluid heat transfer research papers published by scholars from Philadelphia University?* | \<A Numerical Study on Convection Around A Suqare Cylinder using AL2O3-H2O Nanofluid\> |
| STARK-PRIME | *Could you provide a list of investigational drugs that interact with genes or proteins active in the epididymal region?* | \<(S)-3-phenyllactic Acid\>, \<Anisomycin\>, \<Puromycin\> |
| | *Search for diseases without known treatments and induce pruritus in pregnant women, potentially associated with Autoimmune.* | \<Intrahepatic Cholestasis\> |
| | *Please find pathways involving the POLR3D gene within nucleoplasm.* | \<RNA Polymerase III Chain Elongation\> |
| | *Which gene or protein associated with lichen amyloidosis can bind interleukin-31 to activate the PI3K/AKT and MAPK pathways?* | \<OSMR\>, \<IL31RA\> |

Table 1: STARK QA examples which involve semi-structured (relational and textual) information.

# 1 Introduction

Natural-language queries are the primary form of how humans acquire information [17, 21, 27]. For example, users on e-commerce sites wish to express complex information needs by combining free-form elements and constraints, such as "*Can you help me find a push-along tricycle from Radio Flyer that's both fun and safe for my kid?*"in product search. Medical scientists may ask questions like "*What disease is associated with the PNPLA8 gene and presents with hypotonia as a symptom?*". Answering such queries is crucial for enhancing user experience, supporting informed decision-making, and preventing hallucination.

To answer such queries, the underlying knowledge can be represented in semi-structured knowledge bases (SKBs) [35, 40, 50], which integrate unstructured data, such as natural language descriptions and expressions (*e.g.,* description of the tricycle), with structured data, like entity interactions on knowledge graphs (*e.g.,* a tricycle "brand" is Radio Flyer). This allows the SKBs to represent comprehensive knowledge in specific applications, making them indispensable in domains such as e-commerce [15], social media [31], and precision medicine [8, 18, 23].

**Limitations of prior works**. Prior works focused on either purely textual queries on unstructured knowledge [12, 14, 20, 24, 25, 29, 53, 55] or structured SQL [59, 59, 60, 60] or knowledge graph queries [2, 4, 7, 13, 16, 45–47, 57, 58], which are limited in the span of knowledge and inadequate to study the complexities of retrieval on SKBs. Recently, large language models (LLMs) have demonstrated significant potential on information retrieval tasks [14, 30, 43, 61]. Nevertheless, it remains an open question of how effectively LLMs can be applied to the challenging retrieval tasks on SKBs. Moreover, the existing works mainly focus mainly on general knowledge, *e.g.,* from Wikipedia. However, the knowledge may commonly come from private sources, requiring retrieval systems to operate on private SKBs. Therefore, there is a gap of how current LLM retrieval systems handle the complex textual and relational requirements in queries that can involve private knowledge.

**Present work**. To address this gap, we present a large-scale Semi-structure retrieval benchmark on Textual and Relational Knowledge Bases (STARK) (Figure 1). The key technical challenge that we solve is how to accurately simulate user queries on SKBs. This difficulty arises from the interdependence of textual and relational information, which leads to challenges in precisely construct the ground-truth answers from millions of candidates. Additionally, ensuring that queries are useful and resembles real-world scenarios adds further complexity to the benchmarking process.

We develop a novel pipeline that simulates user queries and constructs precise ground truth answers using three SKBs built from extensive texts and millions of entity relations from public sources. We validate the quality of queries in our benchmark through detailed analysis and human evaluation, focusing on their naturalness, diversity, and practicality. Furthermore, we incorporate 274 human-generated queries to compare with synthesized queries and enrich the testing scenarios. With STARK, we delve deeper into retrieval tasks on SKBs, evaluate the capability of current retrieval systems, and provide insights for future advancement. Key features of STARK are:

- **Natural-sounding queries on SKBs (Table 1):** Queries in our benchmark incorporate rich relational information and complex textual properties. Additionally, these queries closely mirror the types of questions users would naturally ask in real-life scenarios, *e.g.,* with flexible query formats and possibly with additional contexts.

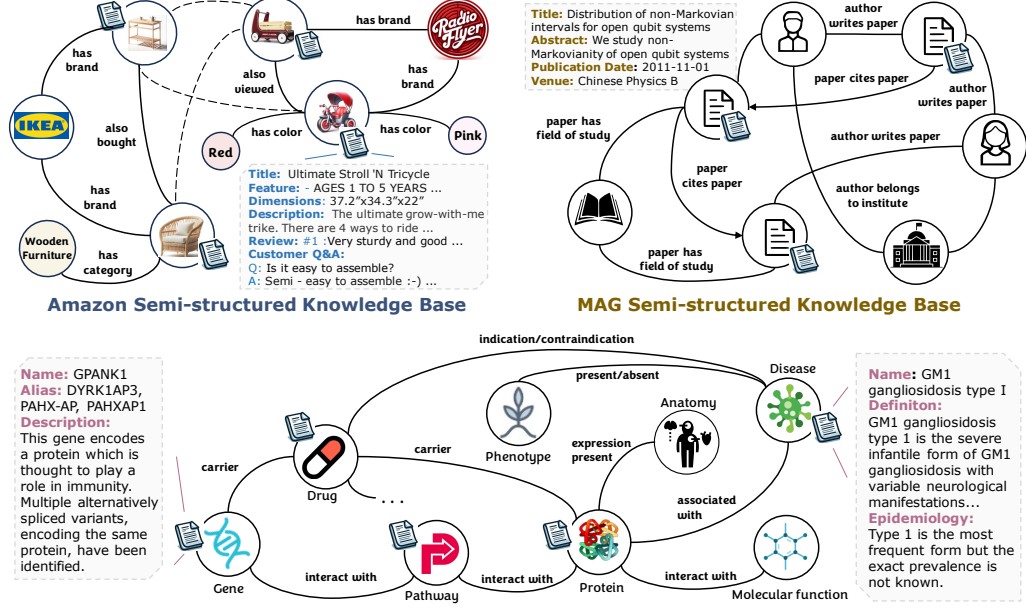

Figure 2: Demonstration of Semi-structured Knowledge Bases, where each knowledge base combines both textual and relational information in a complex way, making the retrieval tasks challenging.

- **Context-specific reasoning:** The queries entail reasoning capabilities specific to the context. This includes the ability to infer customer interests, understand specialized field descriptions, and deduce relationships involving multiple subjects mentioned within the query. For example, the context "*I had a dozen 2.5-inch Brybelly air hockey pucks, so I'm trying to find matching strikers.*" entails the user's interest in looking for complementary products. Such reasoning capabilities are crucial for accurately interpreting and responding to the nuanced requirements of each query.

- **Diverse domains:** Our benchmark spans three knowledge bases[*] for applications including product recommendation, academic paper search, and precision medicine inquiries. STARK provides a comprehensive evaluation of retrieval systems across diverse contexts and domains.

We conduct extensive experiments on LLM retrieval systems, highlighting challenges in handling textual and relational data and latency on large-scale SKBs with millions of entities or relations. Finally, we offer insights into building more capable retrieval systems to handle real-world complexity.

## 2 Benchmarking Retrieval Tasks over Textual and Relational Knowledge

### 2.1 Problem Definition

We are given a Semi-Structured Knowledge Base (SKB), which consists of a knowledge graph $G$ and a collection of free text documents $D$. Formally, let $G = (V, E)$ be the knowledge graph, where $V$ is the set of nodes and $E \subseteq V \times V$ is the set of edges representing relationships between nodes. $D = \bigcup_{i \in V} D_i$ be the collection of free-form text documents associated with the nodes, where $D_i$ is the set of documents associated with node $i$. For example, the product knowledge graph in e-commerce can capture relationships between products and brands/colors/categories, and the corresponding text documents include product descriptions, reviews, *etc*.

We define the tasks on our benchmark datasets as follows: Given the knowledge graph $G = (V, E)$, a collection of free text documents $D$, and a query $Q$, the output is a set of nodes $A \subseteq V$ such that for each node $i \in A$, it satisfies the relational requirements imposed by the structure of $G$ as specified in $Q$, and the associated documents $D_i$ satisfy the textual requirements specified in $Q$.

### 2.2 Semi-structured Knowledge Bases (SKBs)

As shown Figure 2, we construct three large-scale SKBs with the relational and textual information with each entity. See Table 2 for the basic data statistics and Appendix A.1 for details.

---

[*]Explore the SKBs at https://stark.stanford.edu/skb_explorer.html

Table 2: Data statistics of our constructed semi-structured knowledge bases

|  | #entity types | #relation types | avg. degree | #entities | #relations | #tokens |
|---|---|---|---|---|---|---|
| STARK-AMAZON | 4 | 5 | 18.2 | 1,035,542 | 9,443,802 | 592,067,882 |
| STARK-MAG | 4 | 4 | 43.5 | 1,872,968 | 39,802,116 | 212,602,571 |
| STARK-PRIME | 10 | 18 | 125.2 | 129,375 | 8,100,498 | 31,844,769 |

**Amazon Semi-structured Knowledge Base**. The SKB features four entity types: `product`, `brand`, `color`, and `category`, and five relation types: `also_bought`, `also_viewed` between `product` entities, and `has_brand/color/category` associated with the products. We derive the textual information of `product` nodes by combining Amazon Product Reviews [15] with Amazon Q&A Data [32]. This provides a rich amount of texts, including product descriptions and customer reviews. For other entities, we extract their names or titles as the textual attributes. Amazon SKB features an extensive textual data largely contributed from customer reviews and Q&A.

**MAG Semi-structured Knowledge Base**. This SKB includes node entities of `paper`, `author`, `institute`, and `field_of_study`. We derive its relational structure by extracting a subgraph from obgn-mag [19], which contains shared paper nodes with obgn-papers100M [19] and all non-paper nodes. We filter out non-English language papers as we only consider single-lingual queries. The paper documents include their titles and abstracts. Additionally, we integrating details from the Microsoft Academic Graph database (version 2019-03-22) [44, 50], providing extra textual information like paper venue, author and institution names. This SKB demonstrates a large number of relations associate with `paper` nodes, especially on citation and authorship relations.

**Prime Semi-structured Knowledge Base**. We leverage the exisiting knowledge graph PrimeKG [8] which contains ten entity types including `disease`, `gene/protein`, and eighteen relation types, such as `associated_with`, `indication`. Compared to the Amazon and MAG SKBs, Prime SKB is denser and features a greater variety of relation types. While PrimeKG provides text information on `disease` and `drug` entities, we integrate the data from multiple databases for `gene/protein` and `pathway` entities such as genomic position, gene activity summary and pathway orthologous event.

## 2.3 Retrieval Tasks on Semi-structured Knowledge Bases

Our retrieval benchmark (Table 3) consists of three novel retrieval-based question-answering datasets, each comprising synthesized train/val/test sets with 9k to 14k queries in total and a high-quality human-generate query set. The queries synthesize relational and textual knowledge, mirroring real-world queries in terms of natural-sounding property and flexible formats.

**STARK-AMAZON**. The task aims at product recommendation, with a notable 68% of the synthesized queries yielding more than one ground truth answer. The dataset prioritizes customer-oriented criteria, highlighting textual elements such as product quality, functionality, and style. Moreover, it incorporate single-hop relational aspects (Appendix A.2) into the queries, including brand, category, and product connections (*e.g.,* complementary or substitute items). The queries are framed in conversation-like formats, enriching the context and enhancing the dataset's relevance to real-world scenarios.

**STARK-MAG**. Beyond the single-hop relational requirements in STARK-AMAZON, STARK-MAG emphasizes the fusion between the textual requirements with multi-hop queries for precise academic paper search. For example, "Are there any papers from King's College London" highlights the metapath (`institution → author → paper`) on the relational structure. We designed three single-hop and four multi-hop relational query templates (Appendix A.3). The textual aspects focus on the paper's topic, methodology, and contribution *etc.*

**STARK-PRIME**. The task is to answer complex biomedicine inquiries. For synthesized queries, we developed 28 multi-hop query templates (Appendix A.4) to cover various relation types and ensure their practical relevance. For example, the template "What is the drug that targets genes or proteins in <anatomy>?" aids precision medicine by identifying treatments targeted to specific anatomical areas. For `drug`, `disease`, `gene/protein`, and `pathway` entities, the queries are a hybrid of relational and textual requirements. For entities such as `effect/phenotype`, the queries rely solely on relational data due to limited textual information. We exhibit three distinct user roles – medical scientist, doctor, and patient – for generating queries about drug and disease, which diversify the language to comprehensively evaluate the retrieval systems.

Table 3: Statistics on the STARK benchmark datasets.

| | | #queries | #queries w/ multiple answers | average #answers | train / val / test |
|---|---|---|---|---|---|
| Synthesized (Sec 2.4, 2.5) | STARK-AMAZON | 9,100 | 7,082 | 17.99 | 0.65 / 0.17 / 0.18 |
| | STARK-MAG | 13,323 | 6,872 | 2.78 | 0.60 / 0.20 / 0.20 |
| | STARK-PRIME | 11,204 | 4,188 | 2.56 | 0.55 / 0.20 / 0.25 |
| Human-generated (Sec 2.6) | STARK-AMAZON | 81 | 64 | 19.50 | For testing only |
| | STARK-MAG | 84 | 34 | 3.26 | |
| | STARK-PRIME | 98 | 41 | 2.77 | |

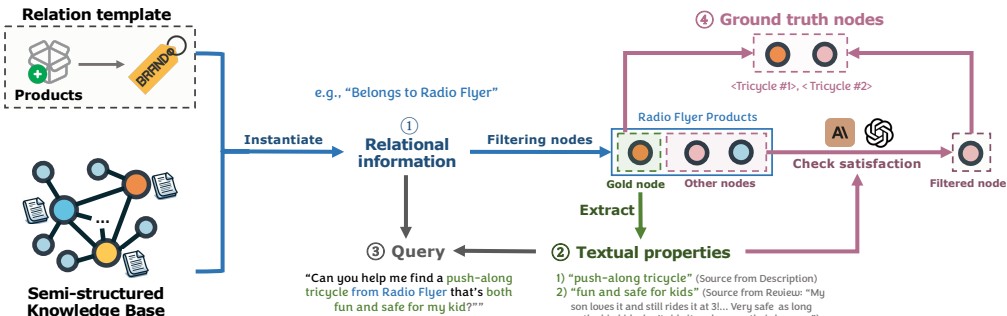

Figure 3: The construct pipeline to generate our semi-structured retrieval datasets.

## 2.4 Benchmark Construction: Synthesized Queries

In Figure 3, we present a novel pipeline that synthesizes the SKB queries and automatically generates the ground truth answers. The key idea is to entangle relational and textual information during synthesis and disentangle them during answer filtering. It involves four steps as follows:

- **1) Sample Relational Requirements:** For each query, we sample a practical relation template constructed with expert/domain knowledge, *e.g.,* "(a product) belongs to <brand>" and ground it with sampled entities (*i.e.,* a specific brand), *e.g.,* "belongs to Radio Flyer". This relational requirement yields a set of candidate entities, *i.e.,* products belonging to Radio Flyer.

- **2) Extracting Textual Properties:** We randomly sample a candidate entity from the first step, referred to as the *gold answer*, from which LLMs extract properties that align with the interests of specific roles (*e.g.,* customers, researchers, or doctors) in its textual document. In Figure 3, we extract multiple properties about the functionality and user experience from a Radio Flyer product.

- **3) Combining Textual and Relational Information:** We use two LLMs to synthesize queries from textual properties and relational requirements, enhancing diversity and reducing bias arise from relying on a single LLM. The first LLM focuses on generating natural, role-specific, and style-consistent (*e.g.,* ArXiv searches) queries. The second LLM enriches the context and rephrases queries, which poses the need for advanced reasoning to comprehend them under complex contexts.

- **4) Filtering Additional Answers:** Finally, we employ multiple LLMs to verify if the candidates from the first step meet the extracted textual properties. Only candidates passing all LLM verifications are included in the final ground truth set. To assess the precision of this filtering mechanism, we compute the average ratios for the gold answers to be verified, which are 86.6%, 98.9%, and 92.3% on the three datasets, highlighting our efficacy in yielding high-quality ground truth answers.

This dataset construction pipeline is automatic, efficient, and broadly applicable to the SKBs in our formulation. We include all of the prompts and the LLMs versions in the above steps in Appendix E.

## 2.5 Synthesized Data Distribution Analysis and Human Evaluation

- **Query and Answer Length**. Query length (in words) reflects the amount of user-provided context information, while the number of answers indicates query ambiguity/concreteness. Figure 4 shows similar query length distributions across the datasets, with most queries around 16 words. Longer queries (up to 50 words) often mention other entities or provide detailed context. Notably, STARK-AMAZON has a significant long-tail pattern, with about 22% of the answers have more than 30 entities, reflecting diverse e-commerce recommendations and ambiguous user queries.

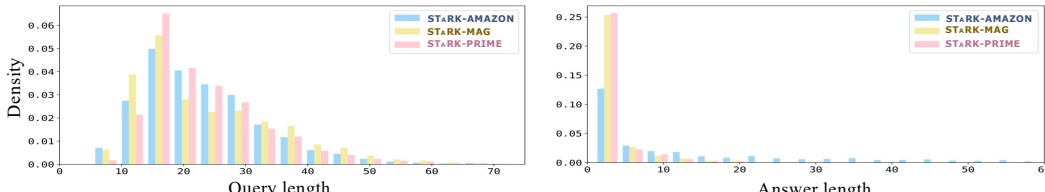

Figure 4: Distribution of query and answer lengths on STARK datasets.

Table 4: Query diversity measurement on STARK. See Appendix B for the metric definition.

Figure 5: Average relative composition of relational vs. textual information.

|  | Shannon Entropy | Type-Token Ratio |
|---|---|---|
| STARK-AMAZON | 10.39 | 0.179 |
| STARK-MAG | 10.25 | 0.180 |
| STARK-PRIME | 9.63 | 0.143 |
| Reference article | 10.44 | 0.261 |

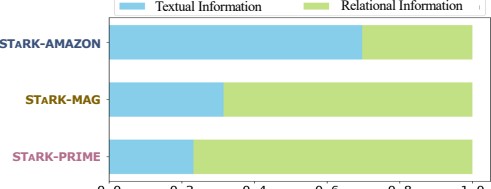

- **Query Diversity**. A diverse set of queries poses challenges for broader applicability to meet varying user demands. We measure query diversity using Shannon Entropy for word distribution and Type-Token Ratio (TTR) for unique words. Higher values indicate greater lexical diversity. Table 4 shows high Shannon Entropy and steady TTR across all datasets. For reference, we compute these metrics for the Wikipedia page of Barack Obama[†].

- **Proportionality of Relational *vs.* Textual Information**. Our benchmark queries feature the composition of textual and relational information. To understand the distribution of information types, we calculate the average ratio of relational to textual requirements by word count in the queries across each dataset. Note that the ratios do not directly reflect their importance in determining final answers. Figure 5 shows varying ratios, which highlights different emphases on textual versus relational requirements and challenges retrieval systems to adapt to different distributions.

**Human evaluation**. We qualitatively assess sampled queries from our benchmark for naturalness (resembling natural conversation), diversity (covering various question structures and complexities), and practicality (relevance to real-world situations) with 63 participants. Evaluation results, converted from a 5-point Likert-like scale to a positive/tie/negative scale, show positive and non-negative rates in Table 10 (Appendix D.1). On average, 94.1%, 85.3%, and 89.4% of participants rated the queries neutral or above in naturalness, diversity, and practicality, respectively. These results validate the quality of our benchmark and its potential for diverse and realistic retrieval tasks.

### 2.6 Benchmark Construction: Human-Generated Queries

To enhance our benchmark's practical relevance, we engaged 31 participants (22 native English speakers) to generate 263 queries across three SKBs following the detailed instructions (Appendix C) along with our interactive platform. We manually verified and filtered the ground truth answers to ensure the answer correctness. Table 3 shows the statistics of the human-generated datasets. Finally, we analyzed the commonalities and differences between synthesized and human-generated queries.

**Commonality**. The number of answers of synthesized and human-generated queries are comparable, indicating a similar level of query ambiguity. Moreover, we observe that most styles of human-generated queries are covered in the synthesized dataset. For example, Table 5 highlights their similarities in short product queries, specific author/field inquiries, and complex contextual queries.

**Difference**. We find that human-generated queries often exhibit more unique expressions compared to synthesized ones, such as "*Give me a **fat cross** and road tire that works with my Diamondback bicycle tube*" and "***this sneaky bone-killing culprit***". This discovery suggests a future direction for our benchmark to incorporate modern and dynamic language nuances.

---

[†]https://en.wikipedia.org/wiki/Barack_Obama

Table 5: Comparison of Human-generated and Synthesized Queries

| Query Type | Human-generated Query | Synthesized Query |
|---|---|---|
| Short and Direct | *Red sweatshirt for proud Montreal Canadiens* | *Suggestions for a Suunto bike mount?* |
| Specific Author & Field | *Find me papers that discuss improving condenser performance authored by Stojan Hrnjak* | *Show me papers by Seung-Hyeok Kye that discuss separability criteria.* |
| Complex Context | *Help me. I am trying to diagnose a patient with persistent joint pain, and I suspect a condition where the bone is dying due to compromised blood supply, often linked to factors like steroid use, ... what's the name of this **sneaky bone-killing culprit**?* | *I'm experiencing joint pain accompanied by swelling... I'm concerned about medications aggravating my fuzzy eyesight and potential blood clotting complications. Could you recommend treatments while minimizing these side effects?* |

## 3 Experiments

### 3.1 Baseline Retrieval Models and Evaluation Metrics

We extensively evaluate five classes of retrieval models described below.

- **Sparse Retriever**: **BM25** [39] is a traditional yet powerful sparse retrieval method based on term frequency-inverse document frequency (TF-IDF). It computes relevance scores by considering the frequency of query terms in documents, adjusted for term rarity and document length.

- **Small Dense Retrievers**: **DPR** [26], **ANCE** [52], and **QAGNN** [56]. These compact models generate dense embeddings for both queries and documents, computing retrieval scores based on embedding similarities. They serve as baselines for comparison with LLM-based dense retrievers.

- **LLM-based Dense Retrievers**: **text-embedding-ada-002 (abbrev. ada-002)** [36], **voyage-large-2-instruct (abbrev. voyage-l2-instruct)** [1], **LLM2Vec-Meta-Llama-3-8B-Instruct-mntp (abbrev. LLM2Vec)** [3], and **GritLM-7b** [33]. These models leverage LLMs to generate dense embeddings that are more contextually expressive.

- **Multivector Retrievers**: **multi-ada-002** [36] and **ColBERTv2** [41]. Beyond ada-002 which represents a document as an embedding, **multi-ada-002** splits each document into overlapping chunks and embeds them using the same encoder as the query. Similarity scores between the query and chunks are aggregated using the average of the top-3 similarities, which we found to perform best. **ColBERTv2** represents each document as multiple token-level embeddings for fine-grained matching, capturing richer semantic information.

- **LLM Rerankers**: **Claude3** and **GPT-4** rerankers [11, 62]. These models improve the precision of top-$k$ ada-002 results by reranking them using large language models. We employ GPT-4-turbo (`gpt-4-1106-preview`) and Claude3 (`claude-3-opus`), setting $k = 20$ for synthesized queries and $k = 10$ for human-generated queries. Given a query, the LLMs assign a satisfaction score from 0 to 1 to each candidate entity based on textual and relational information. Due to high computational costs, we evaluate these rerankers on a random 10% sample of test queries.

The performance of these models are measured using standard retrieval metrics below.

- **Hit@$k$** assesses whether the correct item is among the top-$k$ results from the model. We used $k = 1$ and $k = 5$ for evaluation. At $k = 1$, it evaluates the accuracy of the top recommendation; at $k = 5$, it examines the model's precision in a wider recommendation set.

- **Recall@$k$** measures the proportion of relevant items in the top-$k$ results. For synthesized queries, $k = 20$ is used, as the answer length of all of the queries in our benchmarks are equal or smaller then 20. This metric offers insight into the model's ability to identify all relevant items, particularly in scenarios where missing any could be critical.

- **Mean Reciprocal Rank (MRR)** is a statistic for evaluating the average effectiveness of a predictive model. It calculates the reciprocal of the rank at which the first relevant item appears in the list of predictions. This metric emphasizes the importance of the rank of the first correct answer, which is crucial in many practical applications where the first correct answer is often the most impactful.

Table 6: Testing results on STARK-Syn(thesized).

| | STARK-AMAZON | | | | STARK-MAG | | | | STARK-PRIME | | | |
|---|---|---|---|---|---|---|---|---|---|---|---|---|
| | Hit@1 | Hit@5 | R@20 | MRR | Hit@1 | Hit@5 | R@20 | MRR | Hit@1 | Hit@5 | R@20 | MRR |
| **Full Testing Dataset** | | | | | | | | | | | | |
| BM25 | 44.94 | **67.42** | 53.77 | 55.30 | 25.85 | 45.25 | 45.69 | 34.91 | 12.75 | 27.92 | 31.25 | 19.84 |
| DPR (roberta) | 15.29 | 47.93 | 44.49 | 30.20 | 10.51 | 35.23 | 42.11 | 21.34 | 4.46 | 21.85 | 30.13 | 12.38 |
| ANCE (roberta) | 30.96 | 51.06 | 41.95 | 40.66 | 21.96 | 36.50 | 35.32 | 29.14 | 6.53 | 15.67 | 16.52 | 11.05 |
| QAGNN (roberta) | 26.56 | 50.01 | 52.05 | 37.75 | 12.88 | 39.01 | 46.97 | 29.12 | 8.85 | 21.35 | 29.63 | 14.73 |
| ada-002 | 39.16 | 62.73 | 53.29 | 50.35 | 29.08 | 49.61 | 48.36 | 38.62 | 12.63 | 31.49 | 36.00 | 21.41 |
| voyage-l2-instruct | 40.93 | 64.37 | 54.28 | 51.60 | 30.06 | 50.58 | **50.49** | 39.66 | 10.85 | 30.23 | 37.83 | 19.99 |
| LLM2Vec | 21.74 | 41.65 | 33.22 | 31.47 | 18.01 | 34.85 | 35.46 | 26.10 | 10.10 | 22.49 | 26.34 | 16.12 |
| GritLM-7b | 42.08 | 66.87 | **56.52** | 53.46 | **37.90** | **56.74** | 46.40 | **47.25** | **15.57** | 33.42 | **39.09** | **24.11** |
| multi-ada-002 | 40.07 | 64.98 | 55.12 | 51.55 | 25.92 | 50.43 | 50.80 | 36.94 | 15.10 | **33.56** | 38.05 | 23.49 |
| ColBERTv2 | **46.10** | 66.02 | 53.44 | **55.51** | 31.18 | 46.42 | 43.94 | 38.39 | 11.75 | 23.85 | 25.04 | 17.39 |
| **Random 10% Sample** | | | | | | | | | | | | |
| BM25 | 42.68 | 67.07 | 54.48 | 54.02 | 27.81 | 45.48 | 44.59 | 35.97 | 13.93 | 31.07 | 32.84 | 21.68 |
| DPR (roberta) | 16.46 | 50.00 | 42.15 | 30.20 | 11.65 | 36.84 | 42.30 | 21.82 | 5.00 | 23.57 | 30.50 | 13.50 |
| ANCE (roberta) | 30.09 | 49.27 | 41.91 | 39.30 | 22.89 | 37.26 | 44.16 | 30.00 | 6.78 | 16.15 | 17.07 | 11.42 |
| QAGNN (roberta) | 25.00 | 48.17 | 51.65 | 36.87 | 12.03 | 37.97 | 47.98 | 28.70 | 7.14 | 17.14 | 32.95 | 16.27 |
| ada-002 | 39.02 | 64.02 | 49.30 | 50.32 | 28.20 | 52.63 | 49.25 | 38.55 | 15.36 | 31.07 | 37.88 | 23.50 |
| voyage-l2-instruct | 43.29 | 67.68 | 56.04 | 54.20 | 34.59 | 50.75 | 50.75 | 42.90 | 12.14 | 31.42 | 37.34 | 21.23 |
| LLM2Vec | 18.90 | 37.80 | 34.73 | 28.76 | 19.17 | 33.46 | 29.85 | 26.06 | 9.29 | 20.7 | 25.54 | 15.00 |
| GritLM-7b | 43.29 | **71.34** | **56.14** | 55.07 | 38.35 | **58.64** | 46.38 | 48.25 | 16.79 | 34.29 | 41.11 | 24.99 |
| multi-ada-002 | 40.85 | 62.80 | 52.47 | 51.54 | 25.56 | 50.37 | **53.03** | 36.82 | 15.36 | 32.86 | **40.99** | 23.70 |
| ColBERTv2 | 44.31 | 65.24 | 51.00 | 55.07 | 31.58 | 47.36 | 45.72 | 38.98 | 15.00 | 26.07 | 27.78 | 19.98 |
| Claude3 Reranker | **45.49** | 71.13 | 53.77 | **55.91** | 36.54 | 53.17 | 48.36 | 44.15 | 17.79 | 36.90 | 35.57 | 26.27 |
| GPT4 Reranker | 44.79 | 71.17 | 55.35 | 55.69 | **40.90** | 58.18 | 48.60 | **49.00** | **18.28** | **37.28** | 34.05 | **26.55** |

Table 7: Testing results on STARK-Human(-Generated).

| | STARK-AMAZON | | | | STARK-MAG | | | | STARK-PRIME | | | |
|---|---|---|---|---|---|---|---|---|---|---|---|---|
| Method | Hit@1 | Hit@5 | R@20 | MRR | Hit@1 | Hit@5 | R@20 | MRR | Hit@1 | Hit@5 | R@20 | MRR |
| BM25 | 27.16 | 51.85 | 29.23 | 18.79 | 32.14 | 41.67 | 32.46 | 37.42 | 22.45 | 41.84 | 42.32 | 30.37 |
| DPR (roberta) | 16.05 | 39.51 | 15.23 | 27.21 | 4.72 | 9.52 | 25.00 | 7.90 | 2.04 | 9.18 | 10.69 | 7.05 |
| ANCE (roberta) | 25.93 | 54.32 | 23.69 | 37.12 | 25.00 | 30.95 | 27.24 | 27.98 | 7.14 | 13.27 | 11.72 | 10.07 |
| QAGNN (roberta) | 22.22 | 49.38 | 21.54 | 31.33 | 20.24 | 26.19 | 28.76 | 25.53 | 6.12 | 13.27 | 17.62 | 9.39 |
| ada-002 | 39.50 | 64.19 | 35.46 | 52.65 | 28.57 | 41.67 | 35.95 | 35.81 | 17.35 | 34.69 | 41.09 | 26.35 |
| voyage-l2-instruct | 35.80 | 62.96 | 33.01 | 47.84 | 22.62 | 36.90 | 32.44 | 29.68 | 16.33 | 32.65 | 39.01 | 24.33 |
| LLM2Vec | 29.63 | 46.91 | 21.21 | 38.61 | 16.67 | 28.57 | 21.74 | 21.59 | 9.18 | 21.43 | 26.77 | 15.24 |
| GritLM-7b | 40.74 | 71.60 | 36.30 | 53.21 | 34.52 | 44.04 | 34.57 | 38.72 | 25.51 | 41.84 | **48.10** | 34.28 |
| multi-ada-002 | 46.91 | 72.84 | **40.22** | 58.74 | 23.81 | 41.67 | **39.85** | 31.43 | 24.49 | 39.80 | 47.21 | 32.98 |
| ColBERTv2 | 33.33 | 55.56 | 29.03 | 43.77 | 33.33 | 36.90 | 30.50 | 35.97 | 15.31 | 26.53 | 25.56 | 19.67 |
| Claude3 Reranker | **53.09** | 74.07 | 35.46 | **62.11** | **38.10** | 45.24 | 35.95 | **42.00** | **28.57** | **46.94** | 41.61 | **36.32** |
| GPT4 Reranker | 50.62 | **75.31** | 35.46 | 61.06 | 36.90 | **46.43** | 35.95 | 40.65 | **28.57** | 44.90 | 41.61 | 34.82 |

## 3.2 Results and Analysis

**Results on synthesized queries.** Table 6 presents the results on both the full synthesized test sets and random 10% samples from these sets. In both cases, **BM25**, despite its simplicity, proves to be a strong baseline, outperforming the dense retrieval models such as **ANCE**. We observe that finetuned **DPR** and **QAGNN**, exhibit insufficient performance. This underperformance is likely due to their relatively small model sizes and the risk of overfitting during training. These issues present challenges in effectively training the models on SKBs, where the entity documents can be hard to differentiate without capturing detailed information.

Among the larger models, **ada-002** benefits from superior pretrained embeddings and significantly outperforms **LLM2Vec** by a large margin. **GritLM-7b** delivers excellent performance, surpassing the ada-002 model overall. In contrast, LLM2Vec underperforms due to its limited context length, which is insufficient for encoding the lengthy documents in the SKBs. For multivector retrievers, we found that **multi-ada-002** generally outperforms ada-002, indicating that using multiple vectors per document enhances retrieval effectiveness. Similarly, fine-grained representation allows **ColBERTv2** to capture subtle semantic nuances between queries and documents, leading to largely improved retrieval accuracy.

However, both GritLM-7b and ColBERTv2 generally underperform compared to the rerankers on the random split, especially in terms of Hit@k metrics. This suggests that while these dense retriever models effectively capture semantic information, they may not fully grasp the nuanced relevance judgments required for top-tier retrieval performance. The rerankers, utilizing powerful LLMs like **GPT-4 (gpt-4-1106-preview)** and **Claude3 (claude-3-opus)**, excel by re-evaluating the top candidates and assigning satisfaction scores based on a deeper understanding of the query and document content. This process allows them to better discern subtle contextual cues and relational

Table 8: Latency (s) of the retrieval systems on STARK.

| | DPR | QAGNN | ada-002 | multi-ada-002 | Claude3 Reranker | GPT4 Reranker |
|---|---|---|---|---|---|---|
| **STARK-AMAZON** | 2.34 | 2.32 | 5.71 | 4.87 | 27.24 | 24.76 |
| **STARK-MAG** | 0.94 | 1.35 | 2.25 | 3.14 | 22.60 | 23.43 |
| **STARK-PRIME** | 0.92 | 1.29 | 0.54 | 0.90 | 29.14 | 26.97 |
| **Average** | 1.40 | 1.65 | 2.83 | 2.97 | 26.33 | 25.05 |

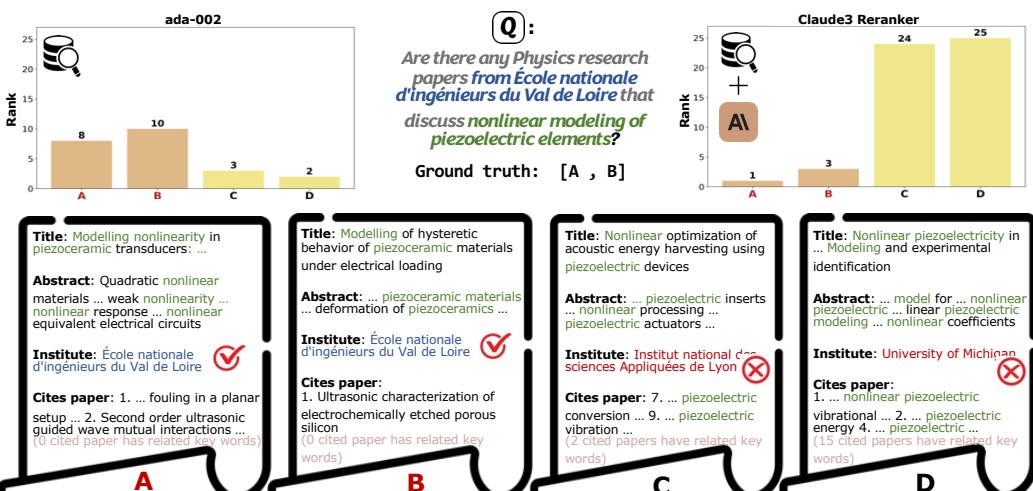

Figure 6: A case study on STARK-MAG shows that ada-002 overranks non-ground truth papers $C$ and $D$ due to repeated keywords in the relational information "cites paper". After reranking with Claude3, it correctly prioritizes ground truth papers $A$ and $B$ with accurate reasoning and analysis.

information that dense retrievers might overlook. Consequently, LLM rerankers enhance retrieval precision at the top ranks.

Finally, regardless of the higher computational costs of the rerankers, their performance remains suboptimal. For instance, the Hit@1 scores for the GPT-4 reranker are only about 18% on STARK-PRIMEand 41% on STARK-MAG, indicating that the top-ranked answers are frequently incorrect. Similarly, the Recall@20 metrics are below 60% across all datasets, with the GPT-4 reranker achieving Recall@20 scores of 55% on STARK-AMAZON, 49% on STARK-MAG, and 34% on STARK-PRIME. This suggests that the ranking results miss a significant portion of relevant answers. The MRR scores are also relatively low, especially for STARK-PRIME, where the GPT-4 reranker attains an MRR of only around 27%.

The insufficient performances may be attributed to the complexity and diversity of queries in SKBs, where nuanced understanding and detailed contextual information are crucial. These findings highlight significant room for improvement in the ranking process.

**Results on human-generated dataset**. Table 7 presents the testing results on the human-generated datasets. For example, the rerankers consistently outperform others, showing their reasoning and context understanding ability. Compared to the synthesized datasets, the performance on human-generated queries is generally higher for most models, but the overall trends remain consistent. This indicates that synthesized datasets may be more challenging, highlighting the complexity of the tasks on our synthesized queries.

Another interesting observation is that the performance of the rerankers is particularly strong on human-generated queries, which may contain more nuanced language and diverse expressions. This suggests that rerankers excel in interpreting and leveraging the richness of human language to improve retrieval accuracy.

**Retrieval latency**. Latency is crucial for practical retrieval systems, as users expect quick responses. As shown in Table 8, we evaluated the latency of various models using a single NVIDIA A100-SXM4-80GB GPU. We observed that the DPR and QAGNN models exhibit lower average latency, making them suitable for time-sensitive applications. In contrast, the ada-002 and multi-ada-002 models have moderate latency due to multiple API calls. However, when combined with LLM rerankers, the

latency increases significantly due to the computational demands of these large models. Therefore, it is important to balance accuracy and latency, especially for complex queries that require advanced reasoning capabilities.

**Case study**. To highlight the importance of reasoning ability for achieving good performance on our benchmark, we present a case study in Figure 6, comparing the ada-002 model with the Claude3 Reranker. In this example, the query requests papers from a specific institution on a particular topic. The ada-002 model fails to address the relational aspect of the query because it embeds entire documents without detailed analysis. This leads to high relevance scores for irrelevant papers that frequently mention keywords like "nonlinear modeling" and "piezoelectric elements" but do not satisfy the relational requirement. In contrast, the LLM reranker significantly improves the results by reasoning about the relationship between the query and each paper, resulting in scores that more accurately reflect relevance. This underscores the need for reasoning ability to grasp query complexities.

## 4    Related Work

**Unstructured QA Datasets**. This research domain consists of methods for retrieving answers from unstructured text, either from a single document [38] or multiple documents [12, 24, 49, 51, 54]. For instance, SQuAD [38] is designed for answer extraction within a specific document, while approaches like HotpotQA [54] and TriviaQA [24] extend to multi-document contexts. Additionally, some studies utilize search engine outputs as a basis or supplementary data for question answering [28, 34]. However, unstructured QA datasets often lack the depth of relational reasoning commonly required in answering complex user queries. In contrast, STARK contains queries demanding multi-hop relational reasoning to challenge model's ability of handling structured information.

**Structured QA Datasets**. These datasets challenge models to derive answers from structured sources such as knowledge graphs [5–7, 13, 16, 47, 58] or tabular data [59, 60]. ComplexWebQuestions [47] and GraphQA [16] propose challenges in interpreting complex queries and textualizing graph structures in KBQA, respectively. For tabular data, WikiSQL [60] focuses on translating queries to SQL for single-table databases, whereas Spider [59] tackles multi-table scenarios. Despite the emphasis on relational data, the restriction to predefined entities and relationships limits the scope of queries. STARK integrates textual content within structured frameworks to enhance the depth and breadth of information retrieval, promoting richer and more nuanced understanding from extensive textual data.

**Semi-Structured QA Datasets**. This category merges tabular and textual data, presenting challenges in semi-structured data comprehension. WikiTableQuestions [37] stresses the integration of table structures with textual elements. TabFact [9], HybridQA [10], and TabMCQ [22] extend this by combining validation of textual statements with tabular reasoning. However, datasets leveraging tables as structured frameworks often lack in depicting the rich relational dynamics among entities. Moreover, prior efforts to link textual and tabular information via external sources have led to cumbersome data constructs. Addressing these challenges, STARK enhances integration, allowing for flexible navigation and advanced retrieval within complex semi-structured knowledge bases, and facilitating more effective relational reasoning and text handling.

## 5    Conclusion and Future Work

We introduce STARK, the first benchmark to thoroughly evaluate LLM-driven retrieval systems for semi-structured knowledge bases (SKBs). Featuring diverse, natural-sounding queries that require context-specific reasoning across diverse domains, STARK sets a new standard for assessing real-world retrieval systems. We contribute three large-scale retrieval datasets with human-generated queries and an automated pipeline to simulate realistic user queries. Our experiments on STARK highlight significant challenges for current models in effectively handling textual and relational information. STARK paves the way for future research to advance complex, multimodal retrieval systems, focusing on reducing retrieval latency and enhancing reasoning abiliites.

Our current SKBs are limited to textual and relational information. Future work should incorporate additional modalities such as images, videos, and speech to provide a more comprehensive information retrieval system. Despite our anonymization efforts, we acknowledge that privacy remains a potential concern when extending this work to other domains with real user data, which should be protected to ensure compliance with privacy regulations.

# 6   Acknowledgement

We thank group members in Leskovec and Zou labs for providing valuable suggestions and conducting benchmark construction. We express our gratitude to the following individuals for their assistance in generating the human-generated queries (ordered by last name):

Michael Bereket, Charlotte Bunne, Yiqun Chen, Ian Covert, Alejandro Dobles, Teddy Ganea, Bryan He, Mika Sarkin Jain, Weixin Liang, Gavin Li, Jiayi Li, Sheng Liu, Michael Moor, Hamed Nilforoshan, Rishi Puri, Rishabh Ranjan, Yanay Rosen, Yangyi Shen, Jake Silberg, Elana Simon, Rok Sosic, Kyle Swanson, Nitya Thakkar, Rahul Thapa, Kevin Wu, Eric Wu, Kailas Vodrahalli.

We especially thank Gavin Li at Stanford University and Zhanghan Wang at New York University for helping build the interactive interface for our SKBs.

We gratefully acknowledge the support of DARPA under Nos. N660011924033 (MCS); NSF under Nos. OAC-1835598 (CINES), CCF-1918940 (Expeditions), DMS-2327709 (IHBEM); Stanford Data Applications Initiative, Wu Tsai Neurosciences Institute, Stanford Institute for Human-Centered AI, Chan Zuckerberg Initiative, Amazon, Genentech, GSK, Hitachi, SAP, and UCB.

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

# A    Benchmark details

## A.1    Semi-structured Knowledge Bases (SKBs)

We present the public sources that we used to construct the SKBs in the table below. We have adhered to the licenses of each public resource.

Table 9: Sources of relational structure and textual information of the benchmarks

|  | relational structure | textual information |
|---|---|---|
| STARK-AMAZON | Amazon Product Reviews | Amazon Product Reviews
Amazon Question and Answer Data |
| STARK-MAG | ogbn-mag | ogbn-papers100M, Microsoft Academic Graph |
| STARK-PRIME | PrimeKG | `disease`: Orphanet; `drug`: DrugBank; `pathway`: Reactome;
`gene`: Ensembl, NCBI Entrez, Uniprot, UCSC, CPDB |

We build an interactive platform to inspect the data of all three SKBs at `https://stark.stanford.edu/skb_explorer.html`. We introduce more detailed data statistics below.

**Amazon SKB**. In total, it comprises around 1.0M entities (`product` entities: 0.9M, `brand` entities: 0.1M, `category` entities: 1.4k, `color` entities: 1.7k) and 9.4M relations (`also_bought`: 2.8M, `also_viewed`: 1.9M, `has_brand`: 1.7M, `has_category`: 2.3M, `has_color`: 0.6M).

**MAG SKB**. This SKB contains around 1.9M entities under four entity types (`author`: 1.1M, `paper`: 0.7M, `institution`: 8.7K, `field_of_study`: 59.5k) and 39.8M relations under four relation types (`author_writes_paper`: 13.5M, `paper_has_field_of_study`: 14.5M, `paper_cites_paper`: 9.7M, `author_affiliated_with_institution`: 2.0M).

**Prime SKB**. The entity count in our knowledge base is approximately 129.3K, with around 8.1M relations. The numbers of entities in each type are listed below:

```
#disease:            17,080
#gene/protein:       27,671
#molecular_function: 11,169
#drug:                7,957
#pathway:             2,516
#anatomy:            14,035
#effect/phenotype:   15,311
#biological_process: 28,642
#cellular_component:  4,176
#exposure:              818
```

## A.2    STARK-AMAZON

**Relational query templates**. These are the **basic** relational templates on STARK-AMAZON. Note that the final relational template can be composed of multiple basic templates. For example, '(color → product ← brand)' represents a relational template combined from two basic relational templates.

| metapath | Query template |
|---|---|
| (brand → product) | "Can you list the products made by <brand>?" |
| (product → product) | "Which products are similar to <product>?" |
| (color → product) | "Can you provide a list of products that are available in <color>?" |
| (category → product) | "What products are available in the <category> category?" |

## A.3    STARK-MAG

**Relational query templates**. We constructed seven relational templates below:

| metapath | multi-hop query template |
|---|---|
| (author → paper) | "Can you list the papers authored by \<author>?" |
| (paper → paper) | "Which papers have been cited by the paper \<paper>?" |
| (field_of_study → paper) | "Can you provide a list of papers in the field of \<field_of_study>?" |
| (institution → author → paper) | "What papers have been published by researchers from \<institution>?" |
| (paper → author → paper) | "What papers have been published by researchers that are coauthors of \<paper>?" |
| (paper → author → paper ← field_of_study ← paper) | "Can you find papers that share a coauthor with \<paper> and are also in the same field of study?" |
| (institution → author → paper ← field_of_study) | "Are there any papers associated with \<institution> and are in the field of \<field_of_study>?" |

For example, the metapath (field_of_study → paper) requires an initial field_of_study entity to be filled in the corresponding query template. For multi-hop metapaths, the last metapath (institution → author → paper ← field_of_study) requires an institution entity and a field_of_study entity to initialize the query.

## A.4 STARK-PRIME

**Relational query templates**. For synthesized queries, we listed 28 multi-hop templates designed by experts to cover various relation types and ensure their practical relevance.

For instance, the query "What is the drug that targets the genes or proteins expressed in \<anatomy>?" serves applications in precision medicine and pharmacogenomics, aiding researchers and healthcare professionals in identifying drugs that act on genes or proteins associated with specific anatomical areas and enabling more targeted treatments.

```
{
(effect/phenotype → [phenotype absent] → disease ← [!indication] ← drug):
    "Find diseases with zero indication drug and are associated with <effect/phenotype>",
(drug → [contraindication] → disease ← [associated with] ← gene/protein):
    "Identify diseases associated with <gene/protein> and are contraindicated with <drug>",
(anatomy ← [expression present] → gene/protein ← [expression absent] ← anatomy):
    "What gene or protein is expressed in <anatomy1> while is absent in <anatomy2>?",
(anatomy → [expression absent] → gene/protein ← [expression absent] ← anatomy):
    "What gene/protein is absent in both <anatomy1> and <anatomy2>?",
(drug → [carrier] → gene/protein ← [carrier] ← drug):
    "Which target genes are shared carriers between <drug1> and <drug2>?",
(anatomy → [expression present] → gene/protein → [target] → drug):
    "What is the drug that targets the genes or proteins which are expressed in <anatomy>?",
(drug → [side effect] → effect/phenotype → [side effect] → drug):
    "What drug has common side effects as <drug>?",
(drug → [carrier] → gene/protein → [carrier] → drug):
    "What is the drug that has common gene/protein carrier with <drug>?",
(anatomy → [expression present] → gene/protein → enzyme → drug):
    "What is the drug that some genes or proteins act as an enzyme upon,
    where the genes or proteins are expressed in <anatomy>?",
(cellular_component → [interacts with] → gene/protein → [carrier] → drug):
    "What is the drug carried by genes or proteins that interact with <cellular_component>?",
(molecular_function → [interacts with] → gene/protein → [target] → drug):
    "What drug targets the genes or proteins that interact with <molecular_function>?",
(effect/phenotype → [side effect] → drug → [synergistic interaction] → drug):
    "What drug has a synergistic interaction with the drug that has <effect/phenotype>
    as a side effect?",
(disease → [indication] → drug → [contraindication] → disease):
    "What disease is a contraindication for the drugs indicated for <disease>?",
(disease → [parent-child] → disease → [phenotype present] → effect/phenotype):
    "What effect or phenotype is present in the sub type of <disease>?",
(gene/protein → [transporter] → drug → [side effect] → effect/phenotype):
    "What effect or phenotype is a [side effect] of the drug transported by <gene/protein>?",
(drug → [transporter] → gene/protein → [interacts with] → exposure):
    "What exposure may affect <drug>s efficacy by acting on its transporter genes?",
```

```
(pathway → [interacts with] → gene/protein → [ppi] → gene/protein):
    "What gene/protein interacts with the gene/protein that related to <pathway>?",
(drug → [synergistic interaction] → drug → [transporter] → gene/protein):
    "What gene or protein transports the drugs that have a synergistic interaction with <drug>?",
(biological_process → [interacts with] → gene/protein → [interacts with] → biological_process):
    "What biological process has the common interactino pattern with gene or proteins as
    <biological_process>?",
(effect/phenotype → [associated with] → gene/protein → [interacts with] → biological_process):
    "What biological process interacts with the gene/protein associated with <effect/phenotype>?",
(drug → [transporter] → gene/protein → [expression present] → anatomy):
    "What anatomy expressesed by the gene/protein that affect the transporter of <drug>?",
(drug → [target] → gene/protein → [interacts with] → cellular_component):
    "What cellular component interacts with genes or proteins targeted by <drug>?",
(biological_process → [interacts with] → gene/protein → [expression absent] → anatomy):
    "What anatomy does not express the genes or proteins that interacts with <biological_process>?",
(effect/phenotype → [associated with] → gene/protein → [expression absent] → anatomy):
    "What anatomy does not express the genes or proteins associated with <effect/phenotype>?",
(drug → [indication] → disease → [indication] → drug)
 & (drug → [synergistic interaction] → drug):
    "Find drugs that has a synergistic interaction with <drug> and both are indicated
    for the same disease.",
(pathway → [interacts with] → gene/protein → [interacts with] → pathway)
    & (pathway → [parent-child] → pathway):
    "Find pathway that is related with <pathway> and both can [interacts with] the same gene/protein.",
(gene/protein → [associated with] → disease → [associated with] → gene/protein)
    & (gene/protein → [ppi] → gene/protein):
    "Find gene/protein that can interect with <gene/protein> and both are associated
    with the same disease.",
(gene/protein → [associated with] → effect/phenotype → [associated with] → gene/protein)
    & (gene/protein → [ppi] → gene/protein):
    "Find gene/protein that can interect with <gene/protein> and both are associated
    with the same effect/phenotype."
}
```

where $[\cdot]$ denotes the relation type.

## B  Mathematical Definitions of Shannon Entropy and Type-Token Ratio

**Shannon Entropy.** Shannon Entropy is a measure of the uncertainty in a set of possible outcomes, quantifying the amount of information or disorder within a dataset. It is defined as follows:

$$H(X) = -\sum_{i=1}^{n} p(x_i) \log p(x_i)$$

where $X$ is the set of possible outcomes, $p(x_i)$ is the probability of occurrence of the outcome $x_i$, and $n$ is the total number of unique outcomes. Higher entropy values indicate greater diversity in the distribution of outcomes.[42]

**Type-Token Ratio (TTR).** The Type-Token Ratio is a measure of lexical diversity, calculated as the ratio of the number of unique words (types) to the total number of words (tokens) in a text. It is defined as follows:

$$\text{TTR} = \frac{V}{N}$$

where $V$ is the number of unique words and $N$ is the total number of words in the text. Higher TTR values indicate a higher proportion of unique words, reflecting greater lexical diversity. [48]

## C  Instructions for Generating Queries

For the process of generating queries by human, the participants were given a list of entity IDs that we randomly sampled from the entire entity set. Then, they were asked to follow the following instructions with the support of our built interactive platform at https://stark.stanford.edu/skb_explorer.html.

> **Task:**
>
> 1) Given the provided entity ID, review the associated document and any connected entities and multi-hop paths.
>
> 2) Find interesting aspects of the entities by examining both their relational structures and the textual information available.
>
> 3) Write your queries from these aspects such that the entity can satisfy all of them.
>
> **Note:**
>
> 1) Please do not leak the name of the entity in the query.
>
> 2) You can skip some entity IDs if you think the knowledge involved is hard to understand.
>
> 3) Feel free to be creative with content of your queries, you can also include additional context. There is NO restriction on how you express the queries.

After collecting the queries, we filtering the ground truth answers manually by human validation.

# D  Experiments

## D.1  More Experimental Results

Table 10: Positive/Non-negative rates (%) from human evaluation.

|  | Naturalness | Diversity | Practicality |
|---|---|---|---|
| STARK-AMAZON | 73.6 / 89.5 | 68.4 / 89.5 | 89.5 / 94.7 |
| STARK-MAG | 94.7 / 100 | 73.7 / 84.2 | 68.4 / 84.2 |
| STARK-PRIME | 67.8 / 92.8 | 71.4 / 82.1 | 71.4 / 89.3 |
| Average | 78.7 / 94.1 | 71.0 / 85.3 | 76.4 / 89.4 |

# E  Prompts and LLM versions for Query Synthesization

We summarize the LLM versions in Table 11. We chose these models based on a joint consideration of their cost, how accurate they are, and whether they were the latest model during different phases of the project. While we used different LLMs, we checked each step separately to make sure the good quality in our benchmark datasets.

Table 11: Summary of LLM Versions for Query Synthesization

| Step | STARK-AMAZON | STARK-MAG | STARK-PRIME |
|---|---|---|---|
| Step 2: Extracting textual requirements | `gpt-3.5-turbo-16k` | `claude-2.0` | `claude-2.0` |
| Step 3: Combining relational and textual requirements | `claude-2.0, gpt-4-0125-preview` | | |
| Step 4: Filtering additional answers | `claude-2.1, claude-2.0, claude-instant-1.2` | | |

## E.1  Extracting textual requirements

> **Prompt for STARK-AMAZON: Textual requirement extraction**
>
> ```
> You are an intelligent assistant that extracts diverse positive
>     ↪ requirements and negative perspectives for an Amazon product.
>     ↪ I will give you the following information:
> - product: <product name>
> ```

```
- dimensions: <product dimensions>
- weight: <product weight>
- description: <product description>
- features: #1: <feature #1> ...
- reviews:
  #1:
    summary: <review summary>
    text: <full review text>
  #2: ...
- Q&A:
  #1:
    question: <product-related question>
    answer: <answer to product-related question>
  #2: ...
Based on the given product information, you need to (1) identify the
    ↪ product's generic category, (2) list all of the negative
    ↪ perspectives and their sources, and (2) extract up to five
    ↪ hard and five soft requirements relevant to customers'
    ↪ interests along with their sources. (1) For example, the
    ↪ product's generic category can be "a chess book" or "a phone
    ↪ case for iphone 6", do not use the product name directly. (2)
    ↪ Negative perspectives are those that the product doesn't
    ↪ fulfill, which come from the negative reviews or Q&A. (3) For
    ↪ the requirements, you should only focus on the product's
    ↪ advantages and positive perspects. Hard requirements mean that
    ↪  product must fulfil, such as size and functionality. Soft
    ↪ requirements are not as strictly defined but still desirable,
    ↪ such as a product is easy-to-use. For (2) and (3), each source
    ↪  is a composite of the key and index (if applicable) separated
    ↪  by "-", such as "description", "Q&A-#1". You should provide
    ↪ the response in a specific format as follows where "item"
    ↪ refers to the product's generic category, e.g., "a chess book".
    ↪
Response format:
{
  "item": <the product's generic category> ,
  "negative": [[<source of negative perspective>, <negative
      ↪ perspective description>]],
  "hard": [[<source of hard requirement>, <hard requirement
      ↪ description>], ...],
  "soft": [[<source of soft requirement>, <soft requirement
      ↪ description>], ...]
}
Here is an example of the response:
{
  "item": "a camping chair",
  "negative": [["reviews-#3", "the chair is not sturdy enough"], ["Q&
      ↪ A-#1", "wrong color"]],
  "hard": [["description", "has a breathable mesh back"], ["
      ↪ description", "the arm is adjustable"], ["dimensions", "more
      ↪ than 35 inches long"], ["features-#7", "with a arm rest cup
      ↪ holder"], ["Q&A-#4", "need to come with a carrying bag"]],
  "soft": [["description", "suitable for outdoors"], ["features-#9",
      ↪ "compact and save space"], ["reviews-#6", "light and portable
      ↪ "]]
}
This is the information of the product that you need to write
    ↪ response for:
<product_doc>
Response:
```



**Prompt for STARK-MAG: Textual requirement extraction**

```
You are a helpful assistant that helps me extract one short
    ↪ requirement (no more than 10 words) about a paper from the
    ↪ paper information that researchers might be interested in. The
    ↪  requirement can be about the paper content, publication date,
    ↪  publication venue, etc. The requirement should be general and
    ↪  not too specific. I will give you the paper information, and
    ↪ you should return a short phrase about the paper, starting
    ↪ with 'the paper...'. This is the paper information:
<doc_info>
Please only return the short and general requirement without
    ↪ additional comments.
```





**Prompt for STARK-PRIME: Textual requirement extraction**

```
You are a helpful assistant that helps me extract <n_properties> from
    ↪  a given <target> information that a <role> may be interested
    ↪ in.
<role_instruction>
Each property should be no more than 10 words and start with "the <
    ↪ target>". You should also include the source of each property
    ↪ as indicated in the paragraph names of the information, e.g.,
    ↪ "details.mayo_symptoms", "details.summary", etc. You should
    ↪ return a list of properties and their sources following the
    ↪ format:
[["<short_property1>", "<source1>"], ["<short_property2>", "<source2
    ↪ >"], ...]
This is the information:
<doc_info>
Please provide only the list with <n_properties> in your response.
    ↪ Response:
```



According to the role assigned to simulate the query content, the `<role_instruction>` as shown below is filled in accordingly.

| role | role instruction |
| --- | --- |
| Doctor | Doctors typically ask questions aimed at diagnosing and treating. Their questions tend to be direct and practical, focusing on aspects involving side effects, symptoms, and complications etc. |
| Medical scientist | Medical scientists often ask questions that reflect the complexity and depth of the scientific inquiry in the medical field. Their questions tend to be detailed and specific, focusing on aspects such as: etiology and pathophysiology, genetic factors, association with pathway, protein, or molecular function. |
| Patient | Patients typically don't know the professional medical terminology. Their questions tend to be straightforward, focusing on practical concerns on the symptons, effects, and inheritance etc., instead of the detailed mechanisms, which may also include more context. |

### E.2   Combining relational and textual requirements

**Prompt for STARK-AMAZON: Fuse relational and textual requirements**

```
You are an intelligent assistant that generates queries about an
    ↪ Amazon item. I will provide you with the item name,
    ↪ requirements, and its negative customer reviews. Your task is
    ↪ to create a natural-sounding customer query that leads to the
    ↪ item as the answer, using the requirements that are non-
    ↪ conflicting with the negative reviews, and provide the indices
    ↪  of the requirements used. For example:

Information:
- item: a soccer rebounder
- requirements:
#1: needs a heavy-duty 1-inch to 3-inch steel tube frame
#2: should be adjustable for practicing different skills
#3: should be durable
#4: usually be viewed together with <SKLZ Star-Kick Hands Free Solo
    ↪ Soccer Trainer>
- negative reviews:
#1: it was broken after a few uses

Response:
{
  "index": [1, 2, 4],
  "query": "Please recommend a soccer rebounder with a steel frame,
      ↪ about 2 inches thick, that can adapt to different skill
      ↪ levels. We had a blast using the <SKLZ Star-Kick Hands Free
      ↪ Solo Soccer Trainer> with my family, and I'm on the lookout
      ↪ for something similar."
}

As the negative review indicates that the soccer rebouncer lacks
    ↪ durability, your query should only incorporate requirements #1,
    ↪  #2, and #4 while excluding #3. A requirement should only be
    ↪ excluded if it conflicts with negative feedback or is unlikely
    ↪  to align with customers' interests. For relational
    ↪ requirements about another <product>, do not directly use "
    ↪ usually bought/viewed together with <product>" in the query.
    ↪ You must deduce the item's relationship with <product> into
    ↪ substitute or complement, and create various user scenarios,
    ↪ such as the item should be compatible or used with <product> (
    ↪ for complements) or match in style with <product> (for
    ↪ substitute), to make the queries sound natural. Except for <
    ↪ product>, you should change the description but convey similar
    ↪  meanings. The query structure is completely flexible. Here is
    ↪  the information to generate the requirement indices and a
    ↪ natural-sounding query:

Information:
<product_req_and_neg_comments>
Response:
```

**Prompt for STARK-MAG: Fuse relational and textual requirements**

```
You are a helpful assistant that helps me generate a new query by
    ↪ incorporating an additional requirement into a given query,
    ↪ and form a coherent and natural-sounding question.
This is the existing query:
```

```
<query>
This is the additional requirement:
<additional_textual_requirement>
You should be creative in combining the existing query and
    ↪ requirement, and flexible in structuring the new query, adding
    ↪  context as needed. Please return the new query without
    ↪ additional comments:
```

The prompt of a second-time rewrite by GPT-4 Turbo:

```
You are a helpful assistant that helps make a researcher's query
    ↪ about a paper more natural-sounding, akin to the language used
    ↪  in ArXiv web searches. You should change the description but
    ↪ convey similar meanings. The query structure is completely
    ↪ flexible. The original query:
"<query>"
Please only output the new query without additional comments:
```

## Prompt for STARK-PRIME: Fuse relational and textual requirements

```
You are a helpful assistant that helps me generate a natural-sounding
    ↪  and coherent query as if you were a <role>. The query should
    ↪ be created based on a list of requirements for searching <
    ↪ plural_target> in a database. I will provide you with the
    ↪ requirements in the following format:
[<requirement1>, <requirement2>, ...]
You should create the query based solely on the given requirements.
    ↪ Moreover, you should craft the query from the perspective of a
    ↪  <role>.
<role_instruction>
For example, a query from a <role> could be
"<example_query>"
You can be flexible in structuring the query and adding additional
    ↪ context. Ensure that the query uses different descriptions
    ↪ than the original property descriptions while retaining
    ↪ similar meanings. The query should sound concise and natural.
    ↪ These are the requirements:
<requirements>
Please create the query based on the given requirements and provide
    ↪ only the query without additional comments. Your response:
```

The prompt of a second-time rewrite by GPT-4 Turbo:

```
You are a helpful assistant that helps me rewrite a query that
    ↪ searches for <plural_target> from the perspective of a <role>.
    ↪  You should maintain the requirements from the original query
    ↪ and the characteristics of the <role>, while being creative
    ↪ and flexible in structuring the query. Ensure the revised
    ↪ query is concise and natural-sounding. Original query: "<query
    ↪ >". Please output only the rewritten query:
```

### E.3 Filtering additional answers

## Prompt for STARK-AMAZON: Filtering additional answers

**Filter products by general category**
```

```
You are an intelligent assistant that identifies whether an Amazon
    ↪ product belongs to a given category. I will give you the
    ↪ product information. You should only answer yes / no in the
    ↪ response. For examples, the product <SKLZ Star-Kick Hands Free
    ↪  Solo Soccer Trainer> belongs to the category "soccer trainer"
    ↪  and the product <Test your Opening, Middlegame and Endgame
    ↪ Play - VOLUME 2> belongs to the category "a chess opening book
    ↪ ", while <Baby Girls One-piece Shiny Athletic Leotard Ballet
    ↪ Tutu with Bow> doesn't belong to category "an adult tutu".

Information:
- product title: <<product_title>>
- product description: <product_description>

Does the product belong to "<target_category>"? Response (yes/no):
```

**Filter products by requirements**

```
You are a helpful assistant that helps me check whether an Amazon
    ↪ product satisfies the given requirements. I will provide you
    ↪ with the product information, which may include the product
    ↪ description, features, reviews, and Q&A from customers. Your
    ↪ task is to assess whether the product meets each requirement
    ↪ based on the provided information. If there is no information
    ↪ that supports the requirement, your response for that
    ↪ requirement is "NA". If there is relevant information that
    ↪ supports the requirement, your response for that requirement
    ↪ is the information source that fulfills the requirement. Each
    ↪ information source is a composite of the key and index (if
    ↪ applicable), separated by "-", such as "description", "
    ↪ features-#3", "Q&A-#1", "reviews-#2". If there are multiple
    ↪ sources,

Response:
{
    1: "NA" or [the information sources that satisfy the requirement
        ↪ #1],
    2: "NA" or [the information sources that satisfy the requirement
        ↪ #2],
    ...
}

Here is the product information:
<product_doc>
The requirements are as follows:
<customer_requirements>

Response:
```

**Prompt for STARK-MAG: Filtering additional answers**

```
You are a helpful assistant that helps me verify whether a given <
    ↪ target_node_type> is subject to a requirement. I will provide
    ↪ you with the <target_node_type> information and the
    ↪ requirement, and you should return only a 'True' or 'False'
    ↪ value, indicating whether the <target_node_type> meets the
    ↪ requirement.
This is the <target_node_type> information:
<doc_info>
```

```
This is the requirement:
<additional_textual_requirement>
Please return only the boolean value without additional comments:
```

## Prompt for STARK-PRIME: Filtering additional answers

```
You are a helpful assistant tasked with verifying whether a given <
    ↪ target> satisfies each of the provided requirements. I will
    ↪ give you the requirements in the following format:
{1: <requirement1>, 2: <requirement2>, ...}
When evidence in the <target> information confirms a requirement is
    ↪ met, cite the source, for example, 'details.mayo_symptoms', '
    ↪ details.summary'. If no direct evidence exists, indicate this
    ↪ with 'NA'. The output in JSON format should be as follows:
{1: 'NA' or <source1>, 2: 'NA' or <source2>, ...}
This is the <target> information:
<doc_info>
These are the requirements:
<requirements>
Please provide only the JSON in your response. Response:
```

