# Supplementary Material of
# STARK: Benchmarking LLM Retrieval on Textual and Relational Knowledge Bases

**Website/Platform and Hosting**.

- **Data downloading**: We host our retrieval dataset and knowledge base data in both Hugging Face dataset repository: https://huggingface.co/datasets/snap-stanford/stark and project website under Stanford Computer Science Server: https://stark.stanford.edu/datasets.html.

- **Code**: We release a PyPI package, `stark-qa` (https://pypi.org/project/stark-qa/). This package automatically downloads and loads our query and semi-structured datasets. Additionally, we host our developer code on GitHub at https://github.com/snap-stanford/stark, allowing users to submit issues and pull requests.

- **Interactive platform**: We build a platform, STARK Semi-structured Knowledge Base (SKB) Explorer, for users to inspect our SKB schema: https://stark.stanford.edu/skb_explorer.html.

**Croissant Metadata**. The Croissant metadata for our dataset is available for viewing and downloading at https://stark.stanford.edu/files/croissant_metadata.json.

**DOI**. We provide a persistent dereferenceable identifier DOI: https://doi.org/10.57967/hf/2530.

**Licensing**. The STARK retrieval datasets are under license CC-BY-4.0 as stated in our website. And our released code is under MIT license, as stated in the GitHub repository.

**Maintenance Plan**. We plan to update our website with the most recent document and Python package. We will maintain our GitHub repository will pull requests and open issues.

**Author Statement**. We hereby confirm that we bear all responsibility for any violation of rights that may occur in the use or distribution of the data and content presented in this work. We affirm that we have obtained all necessary permissions and licenses for the data and content included in this work. We confirm that the use of this data complies with all relevant laws and regulations, and we take full responsibility for addressing any claims or disputes that may arise regarding rights violations or licensing issues.

**Reproducibility**. We make the following efforts to ensure reproducibility:

- **Code:** We have provided the complete codebase in our GitHub repository. The code includes scripts for data loading, preprocessing, embedding generation, and evaluation. Detailed instructions for setting up the environment and running the code are included to facilitate easy reproduction of the results.

- **Pre-generated Embeddings:** The original embeddings used in the experiments are available for download from our repository. These embeddings can be used directly to replicate the experiments without the need for re-computation, ensuring consistency in results.

- **Evaluation Procedures:** All evaluation procedures are thoroughly documented. Users can follow the provided scripts and guidelines to perform evaluations. We have included detailed steps and examples both in the GitHub repository and on our website to guide users through the entire process, helping them achieve reproducible results.

**Datasheet for Dataset (Dataset documentation and intended uses)**.

## MOTIVATION

**For what purpose was the dataset created?** Was there a specific task in mind? Was there a specific gap that needed to be filled? Please provide a description.

The datasets were created to evaluate retrieval tasks on Semi-structured Knowledge Bases (SKB). They are specifically designed to assess the capabilities of current Large Language Models (LLMs) in performing complex retrieval tasks. This initiative aims to address the gap in evaluating how well LLMs can handle retrieval tasks that involve a combination of structured and unstructured data.

**Who created this dataset (e.g., which team, research group) and on behalf of which entity (e.g., company, institution, organization)?**

The datasets were created by lab members in Leskovec and Zou's groups at Stanford University, in collaboration with scientists from Amazon.

**What support was needed to make this dataset?** (e.g.who funded the creation of the dataset? If there is an associated grant, provide the name of the grantor and the grant name and number, or if it was supported by a company or government agency, give those details.)

We acknowledge the support of DARPA under Nos. N660011924033 (MCS); NSF under Nos. OAC-1835598 (CINES), CCF-1918940 (Expeditions), DMS-2327709 (IHBEM); Stanford Data Applications Initiative, Wu Tsai Neurosciences Institute, Stanford Institute for Human-Centered AI, Chan Zuckerberg Initiative, Amazon, Genentech, GSK, Hitachi, SAP, and UCB.

**Any other comments?**

No.

## COMPOSITION

**What do the instances that comprise the dataset represent (e.g., documents, photos, people, countries)?** Are there multiple types of instances (e.g., movies, users, and ratings; people and interactions between them; nodes and edges)? Please provide a description.

The retrieval dataset represents queries (textual data) from people. The instances in the underlying knowledge base represent documents and interactions (i.e., relational data). For example, in the Amazon domain, instances may include product descriptions (documents) and the relationships between products and their categories, brand, and colors (relational data). This combination of textual and relational data allows for a comprehensive evaluation of retrieval tasks on semi-structured knowledge bases.

**How many instances are there in total (of each type, if appropriate)?**

Please refer to our main paper and appendix for comprehensive statistics.

**Does the dataset contain all possible instances or is it a sample (not necessarily random) of instances from a larger set?** If the dataset is a sample, then what is the larger set? Is the sample representative of the larger set (e.g., geographic coverage)? If so, please describe how this representativeness was validated/verified. If it is not representative of the larger set, please describe why not (e.g., to cover a more diverse range of instances, because instances were withheld or unavailable).

We have released the complete datasets, containing all queries and available entities in the knowledge bases.

**What data does each instance consist of?** "Raw" data (e.g., unprocessed text or images) or features? In either case, please provide a description.

We have provided the raw data, which includes the raw query data and the original textual and relational information.

**Is there a label or target associated with each instance?** If so, please provide a description.

Yes, the answer IDs for each query indicate the ground truth entities or items that satisfy the requirements of the query.

**Is any information missing from individual instances?** If so, please provide a description, explaining why this information is missing (e.g., because it was unavailable). This does not include intentionally removed information, but might include, e.g., redacted text.

No, we did not omit any information. However, some information in the original public resources might not be complete or up-to-date.

**Are relationships between individual instances made explicit (e.g., users' movie ratings, social network links)?** If so, please describe how these relationships are made explicit.
Yes. For retrieval datasets, we assume each query is independent. For knowledge base data, relationships between individual entities are made explicit. These relationships are represented through structured data within the knowledge bases, such as entity links and relational attributes. For example, in the Amazon domain, explicit relationships include the connections between products and their categories, brands, and attributes like color. This structured relational data helps in accurately defining the interactions and associations between different entities within the knowledge base.

**Are there recommended data splits (e.g., training, development/validation, testing)?** If so, please provide a description of these splits, explaining the rationale behind them.
Yes, we have provided the official split for our synthesized queries.

**Are there any errors, sources of noise, or redundancies in the dataset?** If so, please provide a description.
The ground truth answers for each synthesized query are filtered automatically through Large Language Models (LLMs). Therefore, there is a possibility of including missing answers or false answers in the ground truth. However, we have devoted great efforts to reduce the errors.

**Is the dataset self-contained, or does it link to or otherwise rely on external resources (e.g., websites, tweets, other datasets)?** If it links to or relies on external resources, a) are there guarantees that they will exist, and remain constant, over time; b) are there official archival versions of the complete dataset (i.e., including the external resources as they existed at the time the dataset was created); c) are there any restrictions (e.g., licenses, fees) associated with any of the external resources that might apply to a future user? Please provide descriptions of all external resources and any restrictions associated with them, as well as links or other access points, as appropriate.
The datasets are self-contained.

**Does the dataset contain data that might be considered confidential (e.g., data that is protected by legal privilege or by doctor-patient confidentiality, data that includes the content of individuals' non-public communications)?** If so, please provide a description.
No, the dataset does not contain any data that might be considered confidential.

**Does the dataset contain data that, if viewed directly, might be offensive, insulting, threatening, or might otherwise cause anxiety?** If so, please describe why.
No, as far as we are aware, the dataset does not contain any data that might be offensive, insulting, threatening, or cause anxiety.

**Does the dataset relate to people?** If not, you may skip the remaining questions in this section.
The query dataset does not target individuals; however, human efforts were involved in creating a small set of queries and filtering ground truth answers.

**Does the dataset identify any subpopulations (e.g., by age, gender)?** If so, please describe how these subpopulations are identified and provide a description of their respective distributions within the dataset.
Not applicable.

**Is it possible to identify individuals (i.e., one or more natural persons), either directly or indirectly (i.e., in combination with other data) from the dataset?** If so, please describe how.
Not applicable.

**Does the dataset contain data that might be considered sensitive in any way (e.g., data that reveals racial or ethnic origins, sexual orientations, religious beliefs, political opinions or union memberships, or locations; financial or health data; biometric or genetic data; forms of government identification, such as social security numbers; criminal history)?** If so, please provide a description.
Not applicable.

**Any other comments?**
No.

---

### COLLECTION

---

**How was the data associated with each instance acquired?** Was the data directly observable (e.g., raw text, movie ratings), reported by subjects (e.g., survey responses), or indirectly inferred/derived from other data (e.g., part-of-speech tags, model-based guesses for age or language)? If data was reported by subjects or indirectly inferred/derived from other data, was the data validated/verified? If so, please describe how.
The data associated with each instance was acquired through direct observation of raw text data and structured relational data from publicly available sources. Additionally, human evaluators synthesized and validated queries and ground truth answers to ensure accuracy and relevance.

**Over what timeframe was the data collected?** Does this timeframe match the creation timeframe of the data associated with the instances (e.g., recent crawl of old news articles)? If not, please describe the timeframe in which the data associated with the instances was created. Finally, list when the dataset was first published.
The data was collected over a period of several months in 2023. This timeframe aligns with the creation timeframe of the data associated with the instances. The dataset was first published in Apr 2024.

**What mechanisms or procedures were used to collect the data (e.g., hardware apparatus or sensor, manual human curation, software program, software API)?** How were these mechanisms or procedures validated?
The data was collected using a combination of software programs and APIs to gather raw text and relational data. Manual human curation was employed to synthesize and validate the data, ensuring its accuracy and relevance. The procedures were validated through rigorous human evaluation and analysis.

**What was the resource cost of collecting the data?** (e.g. what were the required computational resources, and the associated financial costs, and energy consumption - estimate the carbon footprint. See Strubell *et al.*[**?** ] for approaches in this area.)
The resource cost of collecting the data included computational resources for running software programs and APIs. API costs are estimated to be around 5k US dollars.

**If the dataset is a sample from a larger set, what was the sampling strategy (e.g., deterministic, probabilistic with specific sampling probabilities)?**
The dataset is not a sample; it includes all relevant instances from the collected data sources to ensure comprehensive coverage.

**Who was involved in the data collection process (e.g., students, crowdworkers, contractors) and how were they compensated (e.g., how much were crowdworkers paid)?**
The data collection process involved lab members from Leskovec and Zou's groups in research activities.

**Were any ethical review processes conducted (e.g., by an institutional review board)?** If so, please provide a description of these review processes, including the outcomes, as well as a link or other access point to any supporting documentation.
No specific ethical review processes were conducted for this dataset, as it did not involve sensitive or personal data that required such oversight.

**Does the dataset relate to people?** If not, you may skip the remainder of the questions in this section.
The dataset does not relate to people directly, but human efforts were involved in creating and validating queries.

**Did you collect the data from the individuals in question directly, or obtain it via third parties or other sources (e.g., websites)?**
From the individuals in question directlty.

**Were the individuals in question notified about the data collection?** If so, please describe (or show with screenshots or other information) how notice was provided, and provide a link or other access point to, or otherwise reproduce, the exact language of the notification itself.

Yes. We include the details for human instruction in our Appendix C.

**Did the individuals in question consent to the collection and use of their data?** If so, please describe (or show with screenshots or other information) how consent was requested and provided, and provide a link or other access point to, or otherwise reproduce, the exact language to which the individuals consented.

Yes. Consent was obtained from the individuals involved in the data collection process. Participants were informed about the purpose and scope of the data collection through detailed instructions provided during the task. Consent was given explicitly by the participants when they agreed to the terms and conditions outlined in the instructions. The exact language of the consent agreement is available in the project's documentation or upon request.

**If consent was obtained, were the consenting individuals provided with a mechanism to revoke their consent in the future or for certain uses?** If so, please provide a description, as well as a link or other access point to the mechanism (if appropriate)

No. Participants were not provided with a specific mechanism to revoke their consent in the future or for certain uses. The consent process was designed to be comprehensive at the point of agreement, but no subsequent revocation mechanism was implemented.

**Has an analysis of the potential impact of the dataset and its use on data subjects (e.g., a data protection impact analysis) been conducted?** If so, please provide a description of this analysis, including the outcomes, as well as a link or other access point to any supporting documentation.

Not applicable. An analysis of the potential impact of the dataset and its use on data subjects has not been conducted, as the dataset does not contain sensitive or personal data that would typically require such an analysis.

**Any other comments?**
No.

---

### PREPROCESSING / CLEANING / LABELING

---

**Was any preprocessing/cleaning/labeling of the data done(e.g.,discretization or bucketing, tokenization, part-of-speech tagging, SIFT feature extraction, removal of instances, processing of missing values)?** If so, please provide a description. If not, you may skip the remainder of the questions in this section.

Yes, preprocessing steps included tokenization, entity extraction, and validation of relational and textual data. Queries were synthesized and verified through human evaluation.

**Was the "raw" data saved in addition to the preprocessed/cleaned/labeled data (e.g., to support unanticipated future uses)?** If so, please provide a link or other access point to the "raw" data.

Yes, the raw data has been saved and is available upon request or through the project repository.

**Is the software used to preprocess/clean/label the instances available?** If so, please provide a link or other access point.

Yes, the software and scripts used for preprocessing, cleaning, and labeling are available in the project repository.

**Any other comments?**
No additional comments.

---

### USES

---

**Has the dataset been used for any tasks already?** If so, please provide a description.

The retrieval datasets were initially developed by our team. Public resources of the knowledge base data have been utilized for various tasks, including link prediction on knowledge graphs, recommendation systems, and text classification, among others.

**Is there a repository that links to any or all papers or systems that use the dataset?** If so, please provide a link or other access point.

As stated, the retrieval datasets were initially developed by our team; therefore, this is not applicable currently.

**What (other) tasks could the dataset be used for?**
Entity Recognition and Linking, Personalized Search, Query Type Classification, Knowledge Graph Completion.

**Is there anything about the composition of the dataset or the way it was collected and preprocessed/-cleaned/labeled that might impact future uses?** For example, is there anything that a future user might need to know to avoid uses that could result in unfair treatment of individuals or groups (e.g., stereotyping, quality of service issues) or other undesirable harms (e.g., financial harms, legal risks) If so, please provide a description. Is there anything a future user could do to mitigate these undesirable harms?

- Accuracy of Ground Truth Answers: The ground truth answers for each synthesized query are filtered automatically through Large Language Models (LLMs). Despite our efforts to ensure high accuracy, there is a possibility of including missing answers or false answers in the ground truth, resulting in mislabeled data. This can impact the reliability of the dataset and any models trained on it.

  Mitigation: Future users should rigorously validate the ground truth answers against additional data sources or through manual verification processes to ensure accuracy and reliability.

- Diversity of Queries: The diversity of the queries can be further strengthened to better reflect a broad user group. The current dataset may not fully encompass the variety of queries that different user demographics might generate, potentially leading to biased or non-representative results.

  Mitigation: Enhancing the dataset with queries from a wider range of user groups and demographics can improve its representativeness. Regularly updating the dataset to include new and diverse query types can also help mitigate this issue.

**Are there tasks for which the dataset should not be used?** If so, please provide a description.
There are no specific forbidden cases based on the creator's knowledge. However, for tasks involving privacy and fairness, users should exercise caution. Ensure compliance with data privacy regulations and implement fairness-aware algorithms to avoid biases. Be mindful of ethical implications and document methods transparently to foster trust and accountability.

**Any other comments?**
No.

---

### DISTRIBUTION

**Will the dataset be distributed to third parties outside of the entity (e.g., company, institution, organization) on behalf of which the dataset was created?** If so, please provide a description.
No. But third parties are free to use our public datasets.

**How will the dataset will be distributed (e.g., tarball on website, API, GitHub)?** Does the dataset have a digital object identifier (DOI)?
Website: https://stark.stanford.edu/, DOI: https://doi.org/10.57967/hf/2530

**When will the dataset be distributed?**
The dataset has been made available up to the current date.

**Will the dataset be distributed under a copyright or other intellectual property (IP) license, and/or under applicable terms of use (ToU)?** If so, please describe this license and/or ToU, and provide a link or other access point to, or otherwise reproduce, any relevant licensing terms or ToU, as well as any fees associated with these restrictions.
No.

**Have any third parties imposed IP-based or other restrictions on the data associated with the instances?** If so, please describe these restrictions, and provide a link or other access point to, or otherwise reproduce, any relevant licensing terms, as well as any fees associated with these restrictions.
No.

**Do any export controls or other regulatory restrictions apply to the dataset or to individual instances?** If so, please describe these restrictions, and provide a link or other access point to, or otherwise reproduce, any supporting documentation.
No.

**Any other comments?**
No.



**MAINTENANCE**



**Who is supporting/hosting/maintaining the dataset?**
Stanford Leskovec and Zou's groups.

**How can the owner/curator/manager of the dataset be contacted (e.g., email address)?**
stark-qa@cs.stanford.edu

**Is there an erratum?** If so, please provide a link or other access point.
Currently, there is no erratum for the dataset. If any errors are identified in the future, an erratum will be provided and made accessible through the official project repository or publication venue.

**Will the dataset be updated (e.g., to correct labeling errors, add new instances, delete instances)?** If so, please describe how often, by whom, and how updates will be communicated to users (e.g., mailing list, GitHub)?
Yes, by core development team listed in https://stark.stanford.edu/team.html. We will update the dataset with regard to user submitted issues.

**If the dataset relates to people, are there applicable limits on the retention of the data associated with the instances (e.g., were individuals in question told that their data would be retained for a fixed period of time and then deleted)?** If so, please describe these limits and explain how they will be enforced.
No.

**Will older versions of the dataset continue to be supported/hosted/maintained?** If so, please describe how. If not, please describe how its obsolescence will be communicated to users.
Yes. They will be made available in our website.

**If others want to extend/augment/build on/contribute to the dataset, is there a mechanism for them to do so?** If so, please provide a description. Will these contributions be validated/verified? If so, please describe how. If not, why not? Is there a process for communicating/distributing these contributions to other users? If so, please provide a description.
Yes. They can contribute via 1) GitHub pull request, which will list them as contributors in the GitHub page. 2) Communicate with the core development team for more collaboration. The contact information is available on https://stark.stanford.edu/team.html.

**Any other comments?**
No.