# OpenReview forum: "STaRK: Benchmarking LLM Retrieval on Textual and Relational Knowledge Bases"
_NeurIPS.cc/2024/Datasets_and_Benchmarks_Track — NeurIPS 2024 Track Datasets and Benchmarks Poster_

### Official Review · Reviewer_CdN8 · 2024-07-19
**Good and helpful dataset proposal but the results and its availability raise some concerns.**

**Rating:** 5
**Confidence:** 4

**Review:**

The way to generate the synthetic queries and answers is novel and seems to be well designed to leverage existing knowledge bases of structured information. The use of multiple LLMs to generate answers is a good idea to improve diversity.

The experiments do not seem as strong though. The paper doesn’t include results with sparse models and only one dense retrieval model was used. The experimental results are somewhat confusing. One can interpret that “Hit@k” is the equivalent of Precision@k but it would be more instructive to report precision and recall at the same “k” instead of having Hit@1 and H@k with recall@20.

The best performance is achieved with retrieval methods based on LLM reranking but, given that the data was generated synthetically, makes me wonder if there is some intrinsic bias in the model to perform well with these query/answer pairs. If the same models were used in generation and in retrieval, it feels to me that retrieval based on LLM reranking of the synthetic data may not be a fair test. The authors seem to disentangle this by using human generated queries in their analysis. But those results are not shown in the main body of the paper and, more importantly, the differences between models in absolute terms are not as stark as the results with synthetic data. Moreover, they do not include results with dense retrieval models in the human generated data, which could be a good way to gauge if there are biases induced by the LLMs in their synthetic data evaluation.

**Strengths:**

Setting an evaluation task that combines unstructured and structured data is a welcome addition and bridges a gap in retrieval system evaluation. The use of 3 semi-structured knowledge bases seems quite useful.

**Additional Feedback:**

Why not running experiments using dense (or sparse) retrieval models? A set of baseline experiments using BM25 would provide some interesting baseline. For dense retrieval, there are several open source or APIs available to run retrieval experiments. If the intent was to use models that are based on LLMs exclusively, models like LLM2Vec or GritLM seem like viable options. Especially because those models rank high in the MTEB leaderboard. Using multivector retrieval is welcome and appreciated, especially because there seems to be very promising results and the hybrid nature of the dataset makes it a natural fit to multivector. Despite the fact that there are not very many multivector models that leverage the LLMs, it would be great to have a comparison with prior multivector models such as ColBERT or XTR.

**Clarity:**

The paper is clearly written and well structured. The tables and figures are clear and illustrative. However, there are details that the authors are not explicit about and that makes the reader wonder about the quality of the data. For example, what are the specifics of the LLMs used to generate answers? What are the specifics about the ones used to validate the query/answer pairs?

There are other parts of the paper where there seems to be not enough clarity. For example, in their experimental setup they do not describe how the chunks were defined in multivector retrieval. One could assume that the embeddings for the chunks were generated using the setup as in single vector, but that is not clear in the paper. Also, the aggregation method in multivector could be clearer.

**Correctness:**

The formulation of the problem seems correct. The challenging nature of the dataset, including multi-hop queries, is appreciated and should be useful to gauge the reasoning capabilities of the question answering models.

The analysis of the synthetic query validation based on naturalness and diversity seems correct and principled. I am not too sure about “practicality”. Its definition: “relevance to real-world situations”, may have different interpretations depending on cultural context and background.

**Documentation:**

I am confused about what data is being released (if any). The checklist says that the data is proprietary but in the github there are examples of downloading the data to run evaluation.

If the evaluation datasets are proprietary, there is very limited contribution to the research community here. In my opinion, the human generated dataset, despite being small in size, would be a great contribution to the community as it could be used as the starting point for others to build on top of it. If only the training datasets are proprietary, I think it is less than desirable but better. In any case, this should be well explained in the paper.

**Ethics:**

No questions or concerns.

**Limitations:**

Not making the dataset fully available is a limitation of the potential contribution here.

**Opportunities For Improvement:**

In the text there seems to be the implication that question answering is an operation that is intimately related to generative LLMs. For example: “Answering such queries is crucial for enhancing user experience, supporting informed 25 decision-making, and preventing hallucination.” When in fact question answering can be interpreted as an information retrieval operation.

Regarding the construction of the dataset, I wonder if the authors considered using their humana-generated queries as “seed” for the synthetic ones. It feels like that could add even more diversity to the synthetic set.

**Relation To Prior Work:**

The authors could have evaluated the retrieval performance of prior sparse and dense retrieval models on this dataset.

**Summary And Contributions:**

They present a benchmarking dataset that mixes relational and textual and relational knowledge bases that covers domains related to search for products and scientific applications. They develop a pipeline to synthetically generate the queries in this dataset. The quality of the queries is checked by humans. They also add actual human queries to the dataset. They run retrieval experiments on this dataset using different models.

---

> ### Author Rebuttal · Authors · 2024-08-17
>
> We are grateful for your detailed and thoughtful feedback! To address your concerns, we provide point-to-point responses below:
>
> ---
>
> ### **Data availability claim**
>
> We apologize for the oversight in the checklist that caused the confusion. In the supplementary material, we provided a detailed explanation stating that all QA and SKB data are fully available. We have since corrected the checklist information.
>
> ---
>
> ### **Comment batch 1: Experiments setup and baseline details**
>
> - **More Baselines**:
> We now have 9 sparse and dense retrieval models, including the suggested BM25, LLM2Vec, GritLM, and ColBERT, please refer to the PDF in the general response. We also provide more analysis on the results (please refer to our response to `reviewer #1`.
> Besides, all baseline results on the human-generated dataset are also provided to reflect the data quality.
> - **Metric selection between Hit@ and Precision@k**: We introduced the metrics in Appendix D.1. Specifically, Hit@k is a binary measure, being 1 if at least one relevant item is present among the top k results, and 0 otherwise. While Precision@k considers the number of the top k items actually relevant.
> We chose Hit@k over Precision@k because our dataset includes queries of varying lengths. For single-answer queries, Precision@k can only range from [0, 1/k]. However, queries with more than k relevant answers allow Precision@k to range from [0, 1], which may not fairly represent the model's performance across different query types. Hit@k offers a more consistent measure when k is small. Nevertheless, we are happy to provide Precision@20 metric too, if you think that would be a good reference.
> - **MultiVSS details**: This baseline works by splitting a document into smaller chunks, which are defined as segments of the input text with a specified chunk size, and each chunk may overlap with the next by a portion (20%) of the chunk size to capture context across boundaries. The query and each chunk are then embedded using the same encoder. To determine similarity between the query and the document, we aggregate the similarities between the query embedding and each chunk embedding using a function like max, average, or the average of the top k similarities. We reported the results using the average of the top-3 similarities, which is observed to perform better than the max or average aggregation. We added the details to our paper.
>
> ---
>
> ### **Comment batch 2: Dataset construction**
>
> - **What are the LLMs used for different stages of dataset construction?**
> We included a summary of LLM versions for query synthesization in Appendix E (Table 11). We moved the table to the uploaded pdf for your convenience. Specifically, the LLMs in the first two rows are used to generate the queries. Note that we used two different LLMs in `Step 3: Combining relational and textual requirements` to rewrite the querie, and three LLMs to validate the answer.
> - **”How fair are the LLM ranker methods if the LLM models used for generation and retrieval are the same:”** From the previous point, we have used **different** models for generation and retrieval, the models differ in terms of versions and numbers. We believe using multiple LLMs largely alleviate the intrinsic bias, as the information is hard to be decomposed by a single model.
>
> ---
>
> ### **Comment 3: Generative question answering and retrieval**
> In the paper, we state "Our retrieval benchmark consists of three novel retrieval-based question-answering datasets" (L109), which focus on queries targeting specific entities in the knowledge graph. We will emphasize our focus on retrieval-based QA tasks in the introduction to make it clear.
>
> Moreover, as a future direction, the generative-style QA, can be built upon this retrieval-based approach. For instance, a query could evolve from asking "what products" to "what features of the products that...," incorporating both a retrieval step identifying relevant products and a generative step describing their features.
>
> ---
>
> ### **Comment 4: Human-generated query as seed**
> This is a great idea! We think human-generated queries will be a great addition to increase query types. One direct way to integrate them is to provide them as few-shot examples in textual requirement extraction, such that the target on textual requirement can align with users' particular interests better. We added it to our future work!
>
> ---
>
> ### **Comment 5: “Definition of practicality may have different interpretations depending on cultural context and background.”**
> Thank you for bringing this up! In our evaluation, participants are instructed to assess the practicality of queries by considering their relevance to real-world use cases. For example, in the biomedicine domain, queries asking about genes within a very specific range of positions on a chromosome might not be considered practical because such detailed queries may not align with typical real-world needs.
>
> Although we acknowledge that "practicality" can have varied interpretations, we believe such a high level of practicality is still valuable to provide a reference for data quality. Future work can further improve the objectivity of the practicality measure.
>
> ---
>
> ## **Summary**
> We hope our responses effectively address your major concerns, including: 1) code accessibility clarification, 2) more experimental results and details, and 3) clarification on dataset construction and quality.
>
> In our rebuttal, we have comprehended our benchmark by 5 additional dense and sparse retrievers. We acknowledge that the initial submission may have been concise in its description of the experimental setup and baseline implementations, and we have thoroughly revised them to provide greater detail. Additionally, we have discussed potential future improvements, including the use of human-generated queries to increase query diversity.
>
> Therefore, we kindly hope you could reconsider our work in light of the revisions. Thank you for your valuable insights!

---

> > ### Author Response · Authors · 2024-08-27
> >
> > Dear Reviewer #3 CdN8,
> >
> > As the discussion period for our submission is nearing its end, would you mind reading our response and letting us know if your concerns are addressed? Thank you so much!
> >
> > Authors of STaRK

---

> > ### Author Response · Authors · 2024-09-01
> >
> > Dear Reviewer #3 CdN8,
> >
> > We have worked diligently to address your concerns and sincerely hope to receive your reevaluation during the final hours of the discussion phase. Your suggestions have greatly improved our work, for which we are truly grateful.
> >
> > We understand that you might have been busy. We apologize for any hassle caused by our reminders.
> >
> > Thank you so much!
> >
> > Authors of STaRK

---

### Official Review · Reviewer_bote · 2024-07-23
**Innovative approach for benchmarking LLM retrieval**

**Rating:** 9
**Confidence:** 2
**Clarity:** The paper is well written.

**Review:**

Pros:
* This is the first study to utilize semi-structured databases in the context of information retrieval using LLMs, proposing a benchmark and verifying its effectiveness across three different domains.
* The technical contribution is significant in that the use of semi-structured databases enables explicit handling of both textual information and relational information.
* Comparisons with human-generated datasets are also conducted, enhancing the reliability of the results.
* Sufficient information is provided in the appendix, particularly query templates, which will contribute to ensuring reproducibility for readers.

Cons:
* The process of constructing the benchmark using semi-structured databases and LLMs is described in detail, making it relatively easy to understand. However, in the testing scenario, it was difficult to fully comprehend how the decomposition of information for mapping new user queries to the semi-structured database is performed.
* While the paper suggests that the LLM Retrieval system retrieves necessary information from the semi-structured database, more details on the specific technical content of this process would be desirable.

**Strengths:**

The process of constructing the benchmark dataset is described clearly and systematically, with sufficient information provided, including in the appendix. By utilizing semi-structured databases, it seems to adequately demonstrate the usefulness of LLMs as information retrieval tools.

**Additional Feedback:**

I don't have any additional feedback.

**Correctness:**

I had imagined that the claim of using LLMs as information retrievers would involve processes such as decomposition of information to map user queries to the semi-structured database. However, I couldn't find descriptions of such processes. Since the construction of the benchmark seems to be at the forefront, it might be appropriate to modify the main claims somewhat.

**Documentation:**

There is sufficient detail on data collection and organization.

**Ethics:**

I don't see any major ethical concerns that warrant serious further discussion or review.

**Limitations:**

Based on my review of the paper, the authors do not appear to have explicitly addressed limitations or potential negative societal impacts of their work in detail.

**Opportunities For Improvement:**

In the testing scenario, I couldn't clearly understand the technical process of how user queries are processed to access the corresponding knowledge base.

**Relation To Prior Work:**

Yes, the paper does clearly discuss how this work differs from previous contributions.

**Summary And Contributions:**

This study provides a comprehensive benchmark, STARK, by exploiting a semi-structured knowledge base that is more expressive than conventional knowledge graphs. The evaluation includes three different domains, offering compelling generalizability for future usage in more private datasets in the context of information retrieval using LLMs.

---

> ### Author Rebuttal · Authors · 2024-08-17
>
> We appreciate your approval and useful feedback! Here we provide extensive discussion and details to address your question:
>
> ---
> ### **Comment 1: Decomposition of information for mapping new user queries to the semi-structured database in the testing scenarios**
> Thanks for the question!
>
> For multivector retrieval methods such as MultiVSS and Colbert, the decomposition is performed by mapping query information to specific parts of the document. These parts can include feature descriptions of a product explicitly mentioned in the query or certain relations about an entity within the textualized relational information (presented as a sequence of triplets). This allows the retrieval system to focus on the most relevant sections of the document based on the query.
>
> For LLM reranker methods, the decomposition occurs during the process of chain-of-thought prompting or when generating a rationale for the given reranker score. Specifically, we can prompt the LLMs to consider different aspects or requirements within the query, requiring them to find supporting evidence for each aspect. For example, a query can be decomposed into a relational requirement “the product comes from <a specific brand>”, two textual requirements ”the product has shiny colors”, “the product is suitable for kids”. The LLM then assigns a higher score to documents that comprehensively address these various aspects, thereby improving the accuracy of the retrieval process.
>
> However, we found that decomposition is particularly challenging due to the diversity of queries and the complexity of the knowledge bases. For example, in the STaRK-Prime knowledge base, there are 18 different relations, some of which—like “interacts with” and “associates with”—can be expressed in similar terms within the query. This makes it difficult to precisely map the query to the corresponding relational template. We believe this challenge is valuable for testing retrieval systems, as real-world queries often contain ambiguity, requiring the system to handle decomposition in a more flexible and nuanced manner.
>
> To provide further clarity, we added a new section titled "Implementation Technical Details" in our revision, which discusses these aspects in more depth. We hope this response gives you a clearer overview of how information is decomposed in our system. Please feel free to reach out if you need more details; we are happy to provide further explanations.
>
> ---
>
>
> ### **Comment 2: Specific technical content of the LLM retrieval system that retrieves information from the semi-structured database**
> Thanks for your inquiry! Our LLM Retrieval system employs a two-stage approach to retrieve information from the semi-structured database:
>
> We first retrieve the top-k most relevant nodes based on embedding similarity, but it may not be accurate. We've implemented a custom interface that allows the LLM to access the full document information of each node. This information comprises relational data (the node's relationships and one-hop neighbors), textual data (any free-form text documents associated with the node), and metadata (information such as node name, type, and other attributes).
>
> We then prompt LLM reranker to extract the relevant information within the context by implicit summarization or information extraction. The information extraction and summarization can be implicitly guided by the provided query. We instruct the LLM to score each node on a scale of 0.0 to 1.0, based on how well it satisfies the query, considering different aspects and requirements within the query.
> This approach allows us to leverage the LLM's advanced natural language understanding capabilities to improve upon traditional retrieval methods, especially for complex queries that require nuanced interpretation of semi-structured data.
>
> We added these details to the dedicated "Implementation Technical Details" Section.
>
> ---
>
> ### **Limitations and potential negative societal impacts**
>
> We appreciate the reviewer's comment on limitations and potential negative societal impacts. Our current semi-structured knowledge base is limited to textual and relational information, which may not fully cover all information modalities. Future work should incorporate additional modalities such as vision, video, and speech to provide a more comprehensive information retrieval system. While our benchmark covers diverse domains, it may not fully represent all possible real-world scenarios, potentially limiting its generalizability.
>
> Regarding potential negative societal impacts, we acknowledge several concerns. Privacy and data protection are significant issues, as our benchmark includes sensitive domains like product search, academic papers, and precision medicine. Despite anonymization efforts, the use of real user data could raise privacy concerns. As with any AI system, there's also a risk of encoding or amplifying existing biases from the data sources, potentially leading to biased retrieval results. This is particularly concerning in sensitive areas like medicine.
>
> We appreciate the reviewer's concern. We will address the limitations and potential impacts more in detail in the paper's discussion and conclusion sections. This will include the privacy concerns, and potential biases, along with current constraints and limitations. We believe this addition will provide a more comprehensive view of our research and its implications.
>
> ---
> We acknowledge that the previous draft may have been concise in its description of the technical content, and we have carefully revised these sections to provide more comprehensive details. We hope the revisions are clear and helpful. Finally, we sincerely appreciate your recognition of the contributions of our work and we are happy to answer more questions. Thank you once again!

---

### Official Review · Reviewer_XAxN · 2024-08-09
**Review of STaRK: Benchmarking LLM Retrieval on Textual and Relational Knowledge Bases**

**Rating:** 6
**Confidence:** 3
**Correctness:** Yes
**Clarity:** Yes

**Review:**

Please find the detailed review of strengths and questions in the following parts.

**Strengths:**

+ The process of constructing the dataset is clear and reasonable.
+ A good idea was proposed to integrate semi-structured databases using LLM, which places higher demands on the inference ability of existing retrieval systems.
+ The case study effectively validated this point.

**Additional Feedback:**

N/A

**Documentation:**

The document should be more detailed.

**Limitations:**

Please refer to the Opportunities For Improvement part above.

**Opportunities For Improvement:**

+ Some baselines are missing, such as the sparse retrieval model and BM25. At the same time, I also want to be concerned about the effectiveness of some commonly used dense retrieval methods like ANCE and TAS-B, whose results may help the paper provide better conclusions.
+ The LLM-based retrieval model was tested on 10% of the data, while the results of other models were tested on the entire test set, which may result in unfair comparisons. It is necessary to consider releasing the results of other models on this 10%. And Table 9 lacks some result items such as the Recall metric and the results of the dense retrieval model.
+ The two bar charts in Figure 6 seem to be reversed.

**Relation To Prior Work:**

Yes

**Summary And Contributions:**

This paper simultaneously utilizes semi-structured databases and rich textual information to construct a retrieval benchmark on datasets from three different domains. The data construction process is automated and effectively utilizes the generation capability of LLM. The experimental results have proposed adjustments to existing retrieval systems and emphasized the need for more semi-structured retrieval systems.

---

> ### Author Rebuttal · Authors · 2024-08-17
>
> Thanks for your constructive suggestions! Below we address your concerns by following the points raised in the `Opportunities for Improvement` section:
>
> ---
>
> ### **Comment 1: More baselines on sparse and dense retrieval models**
>
> Thanks for the comment! Currently we include 9 sparse and dense retrieval models in total, please refer to our updated results in the uploaded PDF (in the general response). We further provide the following observation:
> ```
> In our experiments with the full synthetic datasets, we observed that the fine-tuned RoBERTa models, including DPR (aka. our previous dense retriever) and ANCE, exhibited insufficient performance, which is likely due to the relatively small model size and the risk of overfitting during training. These issues present challenges in effectively training on SKBs, where documents often share similar structural elements. While BM25, though simple, proved to be a strong baseline, outperforming the finetuned RoBERTa models.
>
> Among the large models, GritLM delivered excellent performance; however, it generally underperformed compared to the rerankers (on the random split) in terms of Hit@1 and Hit@5. This observation aligns with our existing discussion, which highlights the need to enhance reasoning abilities to achieve higher retrieval accuracy in the future.
> ```
> ---
>
> ### **Comment 2: Experimental results on sampled synthetic dataset and human-generated dataset**
>
> The experimental results for all baseline models across all four metrics are also available in the uploaded PDF.
>
> For the random 10% subsets, we will release the splits on our GitHub for public access. Regarding the Recall@20 metric on the human-generated dataset, our previous concern was that this metric may downweight queries with more than 20 answers (e.g., if a query has 40 answers, the maximum recall for that instance would be 0.5), as explained in lines 568-569. However, we believe the recall results still provide useful reference data. Thank you for bringing this up!
>
> ---
>
> ### **Comment 3: “The two bar charts in Figure 6 seem to be reversed.”**
>
> We believe the bar plot is correct. Specifically, the ranks (the lower the rank, the more likely they are to be the correct answers) of A and B are reduced after the Claude3 Ranker, which is desirable since they are the ground truths of the query, while C and D are not. Please let us know if this explanation is clear.
>
> ---
>
> ## **Summary**
>
> We thank you for your time and insights! We hope our experiments address your concerns well.
>
> Finally, we would greatly appreciate your support and reconsideration based on our response, which refines the experiments to be more comprehensive and clear. Thank you again!

---

> > ### Comment · Reviewer_XAxN · 2024-08-22
> > **Thanks for the response**
> >
> > I appreciate the authors for their responses, which address most of my concerns. I will raise my score to 6.

---

> > > ### Author Response · Authors · 2024-08-27
> > >
> > > We appreciate Reviewer #1 XAxN for the approval!
> > >
> > > -- Authors of STaRK

---

### Author Rebuttal · Authors · 2024-08-17

# **General response**

---

We truly appreciate the reviewers' efforts and valuable suggestions in reviewing our paper. We are glad that all/most reviewers reached a positive consensus on our work's presentation, motivation, and novelty. We summarize a table on reviewers’ major comments:

(“*” indicates the concerns that we addressed)

| | Reviewer#1 XAxN | Reviewer#2 bote | Reviewer#3 CdN8 | [Action] Clarification | [Actions] that involve paper modification |
|:---|:---|:---|:---|:---|:---|
|Presentation | “The process of constructing the dataset is clear and reasonable.” |  “The process of constructing the benchmark dataset is described clearly and systematically”|”The paper is clearly written and well structured. The tables and figures are clear and illustrative.”| `NA`| `NA`|
|Dataset novelty | “A good idea was proposed to integrate semi-structured databases using LLM”| “This is the first study to utilize semi-structured databases in the context of information retrieval using LLMs..technical contribution is significant”|”setting an evaluation task that combines unstructured and structured data is a welcome addition and bridges a gap in retrieval system evaluation” | `NA`| `NA`|
|Reproducibility and open-source|NA |”Sufficient information is provided in the appendix, particularly query templates” |* ”what data is being released (if any).”| `We clarify that all synthetic and human-generated data, SKB data, and code are fully available.` |`NA`|
|Experimental details|`NA`|* ”the process in testing scenario…more details on the specific technical content of this process (how LLM Retrieval system retrieves necessary information) would be desirable.”| * (a)”Specifics of the LLMs used to generate and validate answers” (b) ”Specific implementation of MultiVSS“ |`RE Reviewer#3: We point to Appendix E for the LLM information`|`RE Reviewer#2 #3(b): We added extensive technical details on the retrieval process in testing scenarios`|
|Experimental Insights| “The case study effectively validated (higher demands on the inference ability of existing retrieval systems)”| NA | NA | `NA`| `NA` |
|Evaluation|* ”Some baselines are missing..BM25…dense retrieval methods like ANCE and TAS-B”|”Comparisons with human-generated datasets are also conducted, enhancing the reliability of the results.”| * Suggestions on adding more baselines including BM25, LLM2Vec, GritLM, Colbert etc.| `NA` | `Numerous baselines (see uploaded pdf) are added for a more comprehensive evaluatiion` |

---


We see the main concerns from the reviewers are more experimental details and baselines. We appreciate these constructive suggestions and conducted actions accordingly.

As last, we would be thrilled to know whether your concerns have been addressed or if you have any follow-up questions!


— Best,

  Authors of Paper 224

---

**Reference list for the uploaded PDF**

[1] Parishad BehnamGhader, Vaibhav Adlakha, Marius Mosbach, Dzmitry Bahdanau, Nicolas Chapados, and Siva Reddy. 2024. LLM2Vec: Large Language Models Are Secretly Powerful Text Encoders. arXiv preprint (2024).

[2] Vladimir Karpukhin, Barlas Oguz, Sewon Min, Patrick S. H. Lewis, Ledell Wu, Sergey Edunov, Danqi Chen, and Wen-tau Yih. 2020. Dense Passage Retrieval for Open-Domain Question Answering. In EMNLP.

[3] Niklas Muennighoff, Hongjin Su, Liang Wang, Nan Yang, Furu Wei, Tao Yu, Aman-
preet Singh, and Douwe Kiela. 2024. Generative Representational Instruction Tuning.
arXiv:2402.09906 [cs.CL]

[4] OpenAI. 2023. OpenAI Embeddings API. [Software].

[5] Stephen E. Robertson and Hugo Zaragoza. 2009. The Probabilistic Relevance Framework: BM25 and Beyond. Found. Trends Inf. Retr. (2009).

[6] Keshav Santhanam, Omar Khattab, Jon Saad-Falcon, Christopher Potts, and Matei Zaharia. 2022. ColBERTv2: Effective and Efficient Retrieval via Lightweight Late Interaction. In NAACL.

[7] Lee Xiong, Chenyan Xiong, Ye Li, Kwok-Fung Tang, Jialin Liu, Paul N. Bennett, Junaid Ahmed, and Arnold Overwijk. 2021. Approximate Nearest Neighbor Negative Contrastive Learning for Dense Text Retrieval. In ICLR.

[8] Michihiro Yasunaga, Hongyu Ren, Antoine Bosselut, Percy Liang, and Jure Leskovec. 2021.QA-GNN: Reasoning with Language Models and Knowledge Graphs for Question Answering.

---

### Decision · Program_Chairs · 2024-09-26

**Decision:**

Accept (Poster)

**Comment:**

This paper constructs a retrieval benchmark from three domains by using semi-structured databases and rich textual information. All reviewers agree that the construction process is clear and reasonable. Using LLMs to integrate semi-structured databases is interesting. During the rebuttal phase, the authors answered the review questions and addressed most concerns. The authors need to revise the paper according to the reviews and discussions.